# Anti-Inflammatory and Antifibrotic Potential of Longidaze in Bleomycin-Induced Pulmonary Fibrosis

**DOI:** 10.3390/life13091932

**Published:** 2023-09-19

**Authors:** Angelina Pakhomova, Olga Pershina, Pavel Bochkov, Natalia Ermakova, Edgar Pan, Lubov Sandrikina, Yulia Dagil, Lena Kogai, Wolf-Dieter Grimm, Mariia Zhukova, Sergey Avdeev

**Affiliations:** 1Laboratory of Regenerative Pharmacology, Goldberg ED Research Institute of Pharmacology and Regenerative Medicine, Tomsk National Research Medical Centre of the Russian Academy of Sciences, Tomsk 634028, Russia; angelinapakhomova2011@gmail.com (A.P.);; 2NPO Petrovax Pharm LLC, Moscow 123112, Russia; bochkovpo@petrovax.ru (P.B.); dagilya@petrovax.ru (Y.D.); 3Department of Dental Medicine, Faculty of Health, Witten/Herdecke University, 58455 Witten, Germany; prof_wolf.grimm@yahoo.de; 4Department of Pulmonology, Sechenov First Moscow State Medical University, Healthcare Ministry of Russia, 8/2, Trubetskaya Str., Moscow 119991, Russia; serg_avdeev@list.ru; 5Pulmonology Research Institute, Federal Medical and Biological Agency of Russia, 28, Orehovyy Bul., Moscow 115682, Russia

**Keywords:** interstitial lung disease, idiopathic pulmonary fibrosis, hyaluronic acid, Longidaze

## Abstract

Idiopathic pulmonary fibrosis (IPF) is one of the most common forms of interstitial lung disease, characterized by progressive parenchymal fibrosis and respiratory failure. In a model of bleomycin-induced pulmonary fibrosis, the antifibrotic and anti-inflammatory activity of Longidaze (Bovhyaluronidase Azoxymer), which contains a conjugate of the hyaluronidase enzyme with a high molecular weight synthetic carrier azoxymer bromide, was investigated. Experiments were conducted in male C57BL/6 mice. Longidaze was administered at different doses by intranasal and intramuscular routes. Histology, hematology, and enzyme-linked immunosorbent assay were used in the study. The use of Longidaze reduced pulmonary fibrosis, as evidenced by an improvement in histopathologic damage to the lungs, a decrease in the area of connective tissue, and the levels of profibrotic factors (TGF-β1, hydroxyproline, collagen I) in lung tissue. In addition, Longidaze inhibited the inflammatory response in pulmonary fibrosis, and decreased the levels of IL-6, TNF-α, and hyaluronic acid in lung tissue and the recruitment of inflammatory cells into lung tissue. The highest therapeutic efficacy was observed with the use of Longidaze at doses of 120 and 1200 U/kg intramuscularly, which was superior to that of the reference drug pirfenidone axunio. The data presented in this study suggest that Longidaze is a new and promising drug for the treatment of IPF that warrants further investigation in patients with fibrotic interstitial lung disease.

## 1. Introduction

Interstitial lung disease (ILD) is a heterogeneous group of diffuse parenchymal processes in the lung characterized by varying degrees of inflammation and fibrosis. Idiopathic pulmonary fibrosis (IPF) is one of the most common forms of ILD (accounting for 20–30% of all ILD cases), and is characterized by progressive parenchymal fibrosis with a poor prognosis [1,2]. Currently, the incidence and prevalence of IPF are increasing worldwide [3,4]. The median survival of patients with IPF is 3–5 years [5]. Fibrogenesis in IPF is known to be mainly mediated by abnormal release of profibrogenic factors such as tumor necrosis factor-α (TNF-α) and interleukin 1β (IL-1β), which promote myofibroblast differentiation/activation, transforming growth factor-beta (TGF-β) [2] and extracellular matrix (ECM) deposition [6]. Levels of the profibrotic cytokine TGF-β directly correlate with the severity of IPF disease [7,8]. TGF-β is believed to induce epithelial–mesenchymal transition and differentiation of fibroblasts into myofibroblasts, which contribute to the excessive synthesis and deposition of ECM proteins in the lung [9,10]. In addition, the concentration of hyaluronic acid (HA), a key component of the ECM, increases in the lung in IPF [11,12]. In IPF, there is an imbalance between the synthesis and degradation of HA [13]. The main enzyme that regulates HA metabolism is hyaluronidase (HD), which remodels the extracellular matrix by cleaving HA into fragments (glucosamines and glucuronic acid) [13,14]. Given the role of HA in the organization of fibrin, fibronectin, and collagen, it is considered one of the biomarkers of fibrosis and a potential target for antifibrotic therapy, in particular for drugs based on HD [15]. The pathogenesis of pulmonary fibrosis is complex and not fully understood, which hinders the development of specific therapies. Current treatment options for IPF remain limited [8,10]. Pirfenidone axunio and Nintedanib esilat, known on the pharmaceutical market for the treatment of IPF, have not shown the expected high antifibrotic activity; they can only slow down the progression of the disease, and due to the risk of serious side effects, their use is limited [16]. It is known that pirfenidone is an orally administered pyridine, however, the direct targets of pirfenidone are unknown [16]. The antifibrotic property of pirfenidone is demonstrated to depend on its ability to inhibit the direct production of profibrotic cytokines and growth factors, such as TGF-β1, basic-fibroblast growth factor, platelet-derived growth factor, IL-1β, and TNF-α in experimental models of pulmonary fibrosis [17].

Several animal models of pulmonary fibrosis are used to study and understand the biological mechanisms of fibrogenesis in IPF and to test the efficacy of therapeutic compounds. In particular, systemic or intratracheal administration of bleomycin, radiation, or intratracheal administration of silica or asbestos are used to induce fibrosis [18]. At the same time, histologic staining and morphometric analysis of lung tissue, and quantitative determination of hydroxyproline and collagen are used to quantify lung fibrosis in animals [18]. Bleomycin injection into the respiratory tract of small rodents is the most commonly used experimental and preclinical model to study the pathophysiology of IPF and to test potential therapeutic compounds [18,19,20]. The histologic picture of lung tissue in animals after bleomycin administration is similar in some histologic features to that observed in patients with IPF [18,21]. Thus, the drugs pirfenidone axunio and Nintedanib esilat first showed therapeutic efficacy in a model of bleomycin-induced pulmonary fibrosis. As pulmonary fibrosis is also an inflammatory disease [22], it is important to explore the role of immune cells in the pathogenesis of the disease.

In 2011, Bitencourt C.S. and colleagues showed that intranasal administration of HD reduced the fibrotic response induced by bleomycin in the lungs of laboratory animals, reducing TGF-β production and collagen deposition [23]. Thus, HD may represent a new and promising approach/agent for the treatment of pulmonary fibrosis. However, despite the demonstrated efficacy of the drug, the use of native HD is largely limited due to the short plasma half-life when administered systemically [24].

The most effective way to stabilize protein molecules in solution is conjugation of enzymes with high molecular weight carriers. Multipoint covalent and electrostatic binding of the enzyme to the polymer multiplies the conformational stability of the protein molecule, and allows the creation of a macromolecular physiologically active complex intended for long-term presence in various organs and tissues and circulation in the bloodstream. Replacing the native enzyme with a stabilized macromolecular carrier allows for the reduction of the total dose of the drug, increases its retention time in the body, and expands the indications for use, while reducing undesirable side effects.

Previously, we investigated the effects of poloxamer hyaluronidase (pHD—a conjugate of Pluronic L31 and hyaluronidase) compared to native HD in a model of bleomycin-induced pulmonary fibrosis in C57BL/6 mice. It was shown that the administration of pHD reduced the content of connective tissue in the lung by more than twofold, and decreased the concentration of TGF-β, hydroxyproline, type 1 collagen, and total collagen in the lung homogenate [15].

In our publication, the Longidaze is a new formulation of hyaluronidase. The present study aims to investigate the potential anti-inflammatory and antifibrotic effects of Longidaze (Petrovax Pharma LLC, Moscow, Russia) containing HD on a bleomycin model of pulmonary fibrosis in mice. The enzyme preparation Longidaze (INN Bovhyaluronidase Azoximer) developed by NPO Petrovax Pharm is successfully used in medical practice today and has no analogues on the world market, which is a conjugate of hyaluronidase with a high molecular weight carrier, a copolymer of N-oxide 1,4-ethylpiperazine and (N-carboxymethyl)-1,4-ethylpiperazinium bromide (INN Azoximer bromide).

The World Health Organization (WHO) assigned an international nonproprietary name (INN) to the Russian product Longidaze in 2015. Longidaze is able to influence the fibrotic process and is already used for various diseases in urology, gynecology, dermatology, etc. [25]. In addition, the efficacy and safety of Longidaze was evaluated in patients with post-COVID-19 syndrome with lung abnormalities. Patients with post-COVID-19 syndrome and lung abnormalities may benefit from treatment with Longidaze, as evidenced by patients showing improvement in forced vital capacity (FVC), pulse oximetry (SpO_(2)_), functional capacity, and dyspnea mMRC score [26].

## 2. Materials and Methods

### 2.1. Animals

Male 8-week-old C57BL/6 mice (Surgical Bio-modelling Department of the Goldberg ED Research Institute of Pharmacology and Regenerative Medicine, Tomsk, Russia) were used in all experiments. Animals were randomly assigned into the experimental groups. All experimental protocols were approved by the animal care and use committee of the Goldberg ED Research Institute of Pharmacology and Regenerative Medicine, Tomsk NRMC (IACUC Protocol No. 204092022). Within this study, 200 mice were used.

### 2.2. Modeling of Experimental Pulmonary Fibrosis

Experimental pulmonary fibrosis was induced by a single intratracheal bleomycin (BLM, Nippon Kayaku Co., Ltd., Tokyo, Japan) administration at a dose 80 μg/mouse in 0.03 mL of 0.9% NaCl, which was slowly instilled in the tracheal lumen [27]. All procedures were performed under anesthesia induced by inhalation of isoflurane using an apparatus for inhalation of anesthesia UGO BASILE model 21050 (UGO BASILE, Comerio, Italy). The introduction of bleomycin was taken for d0. All mice were euthanized on d7 and d21 by CO_2_ overdose.

Mice were cohoused (5–6 mice per cage) and entrained to a reverse 12 h light/12 h dark cycle. During the 21 days, mice had ad libitum access to standard rodent chow.

### 2.3. Study Design and Experimental Groups

Mice were divided into 10 groups (*n* = 20 in a group): group 1—intact control (mice without bleomycin receiving saline solution formed the control group—Intact); group 2—mice treated with bleomycin (pulmonary fibrosis) receiving NaCl intramuscularly (Vehicle i.m.); group 3—mice treated with bleomycin (pulmonary fibrosis) receiving NaCl intranasally (Vehicle i.n.); group 4—pulmonary fibrosis with treatment by pirfenidone (pirfenidone 300 mg/kg p.o.); group 5—pulmonary fibrosis with treatment by Longidaze intramuscularly at a dose of 60 U/kg (LG 60 U/kg i.m.); group 6—pulmonary fibrosis with treatment by Longidaze intramuscularly at a dose of 120 U/kg (LG 120 U/kg i.m.); group 7—pulmonary fibrosis with treatment by Longidaze intramuscularly at a dose of 1200 U/kg (LG 1200 U/kg i.m.); group 8—pulmonary fibrosis with treatment by Longidaze intranasally at a dose of 10 U/kg (LG 10 U/kg i.n.); group 9—pulmonary fibrosis with treatment by Longidaze intranasally at a dose of 30 U/kg (LG 30 U/kg i.n.); group 10—pulmonary fibrosis with treatment by Longidaze intranasally at a dose of 120 U/kg (LG 120 U/kg i.n.) (Appendix A).

Control animals received 0.03 mL of 0.9% NaCl intramuscularly (group 2) and 0.1 mL or intratracheally (group 3). The animals from groups 5–7 received 0.03 mL of Longidaze intramuscularly, and animals from groups 8–10 received 0.1 mL of Longidaze intranasally.

On d7, hematological blood parameters were studied, inflammation in the lungs was histologically evaluated, and levels of IL-6, TNF-α and hyaluronic acid were assessed in the lung homogenate in mice from groups 1–10 (n = 10/group). On d21, hematological blood parameters were examined, inflammation and connective tissue area in the lungs were histologically evaluated, and ELISA levels of TGF-β, type I collagen, hydroxyproline, and hyaluronic acid were assessed in a lung homogenate of mice from groups 1–10 (n = 10/group). The body weight was recorded on d0, d7, d14, and the day the mice were killed.

### 2.4. Reagents

#### 2.4.1. Longidaze

The enzyme preparation Longidaze (INN Bovhyaluronidase Azoximer) was kindly provided for research by the company “NPO Petrovax Pharm LLC” (Moscow, Russia). Longidaze contains hyaluronidase enzyme stabilized on a carrier as an active ingredient. The enzyme is stabilized by conjugation of hyaluronidase with a polymeric carrier, a copolymer derivative of N-oxide 1,4-ethylenpiperazine and (N-carboxymethyl)-1,4-ethylenpiperazine bromide, an analog of Azoximer bromide with a molecular mass of about 40 kilodalton, which has its own pharmacological activity: immunomodulator, antioxidant, detoxifier (Figure 1). The technology used makes it possible to remove as many inactive impurities from the hyaluronidase as possible, thereby increasing its specific activity while maintaining overall activity after conjugation. The international nonproprietary name of the substance of Longidaze preparation is Bovhyaluronidase Azoximer. Bovhyaluronidase Azoxymer has the ability to regulate the concentration of hyaluronic acid and retains the pharmacological properties of azoxymer bromide with chelating, antioxidant, anti-inflammatory, and immunomodulatory activity [25].

Longidaze was administered intramuscularly at doses of 60, 120, and 1200 U/kg, and intranasally at doses of 10, 30, and 120 U/kg in a course according to the following scheme: d2–d6, d8, d10, d12, d14, d16, d18, and d20.

#### 2.4.2. Pirfenidone

Pirfenidone («TOKYO CHEMICAL INDUSTRY CO., LTD», Tokyo, Japan) was used as the reference drug. Pirfenidone was dissolved in 0.5% carboxymethylcellulose solution (vehicle) and was administered intragastrically daily at a dose of 300 mg/kg/day on d2–d6, d8, d10, d12, d14, d16, d18, d20. The volume of administration was determined according to body weight. The pirfenidone dose was selected according to a report published elsewhere [28].

### 2.5. The Study of Blood Parameters

To determine hematological parameters, blood was taken from the tail vein of the tip of the tail (in vivo) into a microvette with K_3_EDTA spraying for a hematological analyzer in a total volume of 0.2 mL. The analysis was performed on the day of sampling on a Mythic 18 Vet hematology analyzer (EU).

### 2.6. ELISA

#### 2.6.1. IL-6, TGF-β, TNF-α Measurements

The concentrations of IL-1β, TNF-α, and TGF-β in the lung homogenate of the right lung were determined by ELISA according to the manufacturer’s instructions (Cusabio Biotech Co., Ltd., Wuhan, China). Sensitivity was >7.8 pg/mL for IL-6, >4.7 pg/mL for TNF-α, and >0.2 ng/mL for TGF-β. Results are expressed in ng/mL for TGF-β, and pg/mL for TNF-α and IL-6.

#### 2.6.2. Measurements of Hyaluronic Acid, Hydroxyproline, Type I Collagen

Hyaluronic acid, hydroxyproline, and type I collagen levels were determined by ELISA according to the manufacturer’s instructions (Cusabio Biotech Co., Ltd., Wuhan, China). Sensitivity was >15.6 pg/mL for hyaluronic acid, >1.95 ng/mL for hydroxyproline, and >0.039 ng/mL for type I collagen. The results are expressed in ng/mL.

### 2.7. Histological Examination of Lung Tissue

#### 2.7.1. Analysis of Lung Tissue

For histological examination, the left lung was fixed in 10% neutral formalin solution, embedded in paraffin, and Section 5 µm thick were prepared. Sections were stained with hematoxylin and eosin. The structure of the lungs, edema, infiltration by inflammatory cells, venous hyperemia, and thickening of the walls of blood vessels and bronchi were assessed. Micropreparations from each animal were analyzed under an Axio Lab.A1 light microscope (Carl Zeiss, MicroImaging GmbH; Göttingen, Germany) at 100- and 400-fold magnification.

#### 2.7.2. Analysis of Pulmonary Inflammation

The degrees of peribronchial and perivascular infiltrates were assessed by the scale of inflammation and quantitative assessment of peribronchial and perivascular mononuclear cells [29]. The analysis was performed blindly. The drugs were coded and peribronchial inflammation was assessed in a blinded manner using a reproducible scoring system. Each tissue slice was assigned a value from 0 to 3. A value of 0 was taken when no inflammation was detected, a value of 1 was when there was a rare encounter with inflammatory cells, a value of 2 was when most of the bronchi or vessels were surrounded by a thin layer (from one to five cells) of inflammatory cells, and a value of 3 was when most of the bronchi and blood vessels were surrounded by a thick layer (more than five cells) of inflammatory cells (Appendix A). Since there were five to seven randomly selected tissue sections per mouse, inflammation scores could be expressed as an average per animal and compared between groups. In addition, cells in the peribronchial and perivascular segments were counted relative to the length of the basement membrane. The total index of bronchial and vascular inflammation was expressed as the number of inflammatory cells/m of the basement membrane.

#### 2.7.3. The Quantification of the Area of Connective Tissue in the Lungs

To quantify the area of connective tissue, the histological slides were stained using the Van Gieson method [30]. Micrographs of the histological surface section of the lung tissue at 4× magnification were taken for each experimental animal using a Cytation 5 multimode reader (BioTek Instruments, Inc., Winooski, VT, USA). The resulting images were processed using Gen5 software (BioTek, Bad Friedrichshall, Germany). The area of connective tissue in the lung was determined using a function for counting the area of the object in the image. Bronchovascular strands were carefully removed from the analyzed areas. The area of connective tissue (X) was calculated by the formula:X = Σ a × 100/(S − Σ b), (1)
where ∑ a is the sum of pixels occupied by fibrosis tissue on 10 images of one slide, S is the number of pixels corresponding to the total area of the image (when using this camera and program–4423680), and b is the sum of pixels occupied by the empty part of the slide, on 10 images of one slide. The content of connective tissue in the lungs was expressed as a percentage of the lung tissue area.

### 2.8. Statistical Analysis

Statistical analysis was performed using SPSS (version 15.0, SPSS Inc., Chicago, IL, USA). Data were analyzed and presented as means ± standard error of the mean. Statistical significance was evaluated by Student’s *t*-test (for parametric data), or Mann–Whitney test (for nonparametric data) when appropriate. A *p*-value of less than 0.05 (by two-tailed testing) was considered an indicator of statistical significance. Statistical analysis of hematology and ELISA data was performed using the Kruskal–Wallis test, post hoc Wilcoxon test for pairwise comparisons with the bleomycin control group (with appropriate route of administration), and Bonferroni correction (ns—*p* > 0.05, * *p* < 0.05, ** *p* < 0.01, *** *p* < 0.001, **** *p* < 0.0001).

## 3. Results

### 3.1. Mortality Assessment

The condition of mice in all experimental groups was satisfactory, and no deaths were observed.

### 3.2. Body Weight of Mice in Formation of Pulmonary Fibrosis

In the groups of mice with pulmonary fibrosis (groups 2 and 3) without treatment, the body weight of the animals was significantly reduced from day 7 to day 21 of the experiment compared with the intact group (Figure 2a). In the group of mice treated with the reference drug pirfenidone (group 4), a significant decrease in body weight was observed from day 7 to day 21 of the experiment compared to untreated animals (Figure 2a). Body weight loss is common with pirfenidone, as it has a metabolic side effect. In mice treated with Longidaze, body weight was significantly higher compared to animals treated with pirfenidone (Figure 2b,c). The most pronounced effect of Longidaze was seen at a dose of 1200 U/kg when administered intramuscularly, and at a dose of 120 U/kg when administered intranasally (Figure 2).

### 3.3. Influence of Longidaze on Hematologic Parameters of Blood

During the inflammatory phase of fibrosis development, on d7 after bleomycin administration, an increase in the white blood cells (WBCs) in the peripheral blood of mice was observed in both pathological control groups (groups 2 and 3) compared to the group of intact animals (Figure 3). A shift in the relative content of blood cells towards neutrophils was observed in these groups. Against the background of Longidaze administration at all doses, regardless of the route of administration, a decrease in WBCs was observed compared to groups 2 and 3, with a shift in the relative content of cells towards lymphocytes. With the intramuscular injection of Longidaze, an increase in the dose of the drug induces an increase in the effect (Figure 3). The most pronounced effect of the drug was observed at a dose of 1200 U/kg administered intramuscularly (group 7). With the inhalation administration of Longidaze, the greatest activity during these periods of the experiment was observed at doses of 10 U/kg (group 8) and 30 U/kg (group 9) (Figure 3). Administration of pirfenidone prevented the increase in blood WBCs, but the effect of Longidaze was more pronounced.

On d21, the inflammatory response was still present in the peripheral blood of mice in the pathological control groups, as evidenced by the increased content of neutrophils and the continued shift in the relative content of blood cells toward neutrophils compared with the intact control (Figure 4). At the same time, a decrease in WBC and lymphocyte content was observed in the blood of animals from these groups compared to the intact control. Administration of pirfenidone to mice with pulmonary fibrosis (group 4) resulted in the development of an even more pronounced leukopenia in the peripheral blood of the animals by d21 of the experiment (Figure 4). During these study periods, administration of Longidaze at a dose of 1200 U/kg (intramuscular administration, group 7) and 120 U/kg (inhalation administration, group 10) normalized the levels of neutrophils and monocytes, although the levels of lymphocytes remained reduced compared to the intact control (Figure 4). The positive effect of the drug Longidaze compared to the reference drug was remarkable: Longidaze prevented the leukopenia that develops after bleomycin administration and increases after pirfenidone administration.

### 3.4. Effects of Longidaze on Bleomycin-Damaged Lungs

Histologic examination showed that bleomycin induced lesions similar to acute fibrosing alveolitis in the lungs of mice. On d7, inflammatory infiltrates of lymphocytes and macrophages in the parenchyma, hyperemia of large vessels, and vessels of the microvasculature were observed (Figure 5). Small hemorrhages were present in the lung parenchyma. The inflammatory response was predominantly perivascular and peribronchial. There was infiltration of the alveolar septa with lymphocytes, plasmocytes, and multiple accumulations of macrophages in the alveoli.

Injection of Longidaze and the reference drug pirfenidone reduced the intensity of the inflammatory reaction in the lung parenchyma. On d7, intramuscular injection of Longidaze in doses of 60, 120 U/kg (groups 5 and 6) and intranasal application of this drug in doses of 30, 120 U/kg (groups 9 and 10) showed a moderate anti-inflammatory effect similar to that of pirfenidone. The severity of hyperemia of the microvascular bed decreased, the prevalence of inflammatory infiltration of the lung parenchyma and the number of macrophages in the alveoli decreased, and inflammatory infiltrates remained mainly peribronchial and perivascular (Figure 5). Intranasal administration of Longidaze at a minimum dose of 10 U/kg (group 8) did not affect the intensity of the inflammatory response. The most pronounced anti-inflammatory effect was observed with the intramuscular administration of Longidaze at a dose of 1200 U/kg on d7 (group 7). In this group of animals, a significant decrease in the intensity of inflammatory infiltration of the lung parenchyma by lymphocytes, plasmocytes, and macrophages and a significant decrease in the cellularity of peribronchial and perivascular infiltrates were observed. Isolated macrophages were observed in the alveolar lumen. There was local hyperemia, and small vessels were filled with blood corresponding to intact animals.

On d21 after bleomycin instillation, there was an increase in inflammatory infiltration, mainly by lymphocytes and fibroblasts, resulting in thickening of the alveolar septa (Figure 6). Most of the alveoli were filled with macrophages. There were areas of parenchyma where the lung pattern was completely absent. An overgrowth of fibrous tissue was observed in the lungs of the mice. Fibrotic foci were localized peribronchially and perivascularly. Extensive collagen deposition was found in the lung parenchyma, with a significant decrease in the airiness of the organ and the formation of a “honeycomb lung”. There were areas of emphysematous enlarged alveoli, atelectasis. Morphological changes in the lungs of animals in groups 2 and 3 were of the same type.

On d21, the most pronounced anti-inflammatory effect was observed in the group of mice treated intramuscularly with Longidaze at a dose of 1200 U/kg (group 7) (Figure 6). A significant decrease in the number of macrophages, lymphocytes, plasmocytes, and fibroblasts was observed in the lung parenchyma, peribronchial, and perivascular infiltrates, and there were no inflammatory cells in the alveoli. Lung tissue aeration was significantly higher than in pathological control animals (groups 2 and 3). The anti-inflammatory effect of Longidaze was somewhat weaker in the groups of animals that received the drug intramuscularly and intranasally at the dose of 120 U/kg (groups 6 and 10); the intensity of the inflammatory infiltrates decreased in these groups, and individual macrophages were found in the alveoli. The nature of the inflammatory response of the lung parenchyma in other groups of mice treated with Longidaze did not differ from the similar parameter in the model groups (Figure 6).

The most effective antifibrotic effect of Longidaze was demonstrated at a dose of 1200 U/kg (group 7) administered intramuscularly. In this group of mice, the area of fibrous tissue approached that of intact animals. A pronounced antifibrotic effect of Longidaze was observed at a dose of 120 U/kg (group 6) administered intramuscularly. The area of fibrous tissue in the lungs of mice in this group decreased compared to the model animals, and focal collagen deposition was noted, mainly around the large bronchi. In groups of mice receiving Longidaze intramuscularly at a dose of 60 U/kg (group 5) and intranasally at doses of 10, 30, and 120 U/kg (groups 8–10), the antifibrotic effect of the corrector was less pronounced, and close to that of the reference drug pirfenidone (group 4).

### 3.5. Effects of Longidaze on Perivascular and Peribronchial Inflammation in the Lungs

On d7 after bleomycin instillation, an accumulation of inflammatory cells was observed in the peribronchial and perivascular areas of the lung tissue in animals of groups 2 and 3 (Figure 7 and Appendix A). Pirfenidone (group 4) had little effect on the severity of the inflammatory response in the lungs of mice during these experimental periods. Intramuscular administration of Longidaze (groups 5–7) caused a decrease in the recruitment of inflammatory cells to the lung tissue compared to group 2, with the effect becoming greater as the dose of the drug increased. The most pronounced effect was observed with intramuscular injection of Longidaze at a dose of 1200 U/kg (group 7) (Figure 7 and Appendix A). The effect of the drug Longidaze with intranasal administration (groups 8–10) was weakly expressed during these periods of the experiment.

On d21 after bleomycin instillation in the lungs of mice of groups 2 and 3, a pronounced inflammatory reaction was still observed in the peribronchial and perivascular spaces of the respiratory tract. With the injection of the reference drug pirfenidone (group 4) on d21, the level of perivascular and peribronchial inflammation decreased compared to groups 2 and 3, and was significantly higher than in intact animals. The effect of intramuscularly administered Longidaze at a dose of 120 U/kg (group 6) was at the level of the reference drug, and at a dose of 1200 U/kg (group 7) was significantly higher than the effect of pirfenidone. Intranasal administration of Longidaze contributed to a decrease in the accumulation of inflammatory cells in the peribronchial and perivascular space of the respiratory tract only at a dose of 120 U/kg (group 10), while the efficacy was comparable to that of the reference drug pirfenidone.

### 3.6. Effects of Longidaze on Pulmonary Connective Tissue Content

Fibrosis modeling caused a significant increase in the area of connective tissue in the lungs of mice in the pathological control groups by d21 (Figure 8a,b). Injection of pirfenidone (group 4) resulted in a twofold decrease in the area of connective tissue in the lungs compared to groups 2 and 3. Treatment of pulmonary fibrosis with Longidaze at doses of 10 U/kg, 30 U/kg, and 120 U/kg (groups 8–10) by intranasal and intramuscular administration at a dose of 60 U/kg (group 5) produced an effect comparable to pirfenidone: the connective tissue area was significantly lower than the corresponding pathological control, while the indicator remained 23–29% higher than in group 4 (Figure 8b–d and Appendix A). Longidaze at doses of 120 U/kg (group 6) and 1200 U/kg (group 7) showed the most pronounced antifibrotic effect when administered intramuscularly. The area of connective tissue in the lungs of animals treated with Longidaze at a dose of 120 U/kg was 50% lower than the corresponding values in group 2, and 13% lower than the reference drug group (group 4) (Figure 8b). In the treatment of pulmonary fibrosis with Longidaze at a dose of 1200 U/kg (group 7), the area of connective tissue in the lungs decreased by 65% compared to group 2, and by 39% compared to the group of animals treated with the reference drug pirfenidone (Figure 8c).

The highest therapeutic efficacy was observed with intramuscular administration of Longidaze at doses of 120 and 1200 U/kg (groups 6 and 7, respectively). Intranasally administered Longidaze showed anti-inflammatory and antifibrotic properties at a dose of 120 U/kg (group 10).

### 3.7. Enzyme-Linked Immunosorbent Assay

#### 3.7.1. Levels of IL-6, TNF-α, and Hyaluronic Acid in Bleomycin-Induced Lung Injury

On d7 after bleomycin instillation in the lungs of mice of both groups of pathological control (groups 2 and 3), an increase in the level of IL-6, TNF-α, and HA was observed in comparison with intact mice (Figure 9). Intramuscular administration of Longidaze at doses of 120 U/kg (group 6) and 1200 U/kg (group 7) contributed to a decrease in the level of IL-6, TNF-α, and HA in the lung tissue of mice on d7 compared to animals from group 2 (Figure 9a). It should be noted that the effect of these doses of Longidaze on the level of IL-6 and HA was comparable to the effect of the reference drug pirfenidone, while pirfenidone did not affect the level of TNF-α in the lung tissue on the d7. The level of TNF-α in the lung tissue on d7 was positively influenced by the drug Longidaze at an inhalation dose of 30 U/kg (group 9). Notably, the inhalation use of Longidaze at all doses studied (groups 8–10) decreased the level of IL-6, and practically did not change the HA level (Figure 9b).

#### 3.7.2. The Concentration of TGF-β1 and Hyaluronic Acid in Bleomycin-Induced Lung Damage on d21

It was found that the modeling of pulmonary fibrosis on d21 led to a significant increase in the levels of HA and TGF-β1 in the lung homogenate obtained from mice of both pathological control groups (Figure 10). Treatment of animals with pirfenidone resulted in a decrease in the levels of HA and TGF-β1 in lung homogenate compared to those in groups 2 and 3 (Figure 10). Treatment with Longidaze at all doses and routes of administration decreased HA and TGF-β1 levels to varying degrees. The most pronounced decrease was observed with intramuscular injection of Longidaze at a dose of 1200 U/kg (group 7) (Figure 10).

#### 3.7.3. The Collagen I and Hydroxyproline Concentration in Bleomycin-Induced Lung Damage on d21

Modeling pulmonary fibrosis caused a significant increase in the levels of fibrosis markers—collagen I (3.7–3.8-fold) and hydroxyproline (4.3-fold)—in the lung homogenate on d21 in animals of groups 2 and 3 (Figure 10). The use of the pirfenidone (group 4) resulted in a 2-fold and 1.2-fold decrease in the concentration of hydroxyproline and collagen I, respectively, in the lung homogenate of mice with pulmonary fibrosis (Figure 10). The concentration of hydroxyproline (2-fold) and collagen I (3-fold) in lung homogenate was most effectively reduced by Longidaze administered intramuscularly at a dose of 1200 U/kg (group 7) (Figure 10a). The effect was weaker at a dose of 120 U/kg with intranasal (group 10) and intramuscular (group 6) administration, but the efficacy of Longidaze even at this dose exceeded that of pirfenidone (Figure 10). Thus, the use of Longidaze at all doses and routes of administration studied reduced the levels of hydroxyproline and collagen I in lung homogenate compared to untreated mice with pulmonary fibrosis (Figure 10). The most pronounced effect, exceeding that of the reference drug pirfenidone, was observed in groups of mice treated with Longidaze intramuscularly at a dose of 1200 U/kg (group 7) and intranasally at a dose of 120 U/kg (group 10) (Figure 10).

## 4. Discussion

IPF is a chronic progressive disease with the development of interstitial fibrosis and progressive respiratory failure [2,31]. In most cases, the prognosis of IPF is unfavorable, with an average life expectancy of about five years from diagnosis [5]. Current therapies, Nintedanib and pirfenidone, slow progression by preserving lung function. However, an adequate drug treatment that would stop the progression of the disease has not been discovered, which requires the development of new effective antifibrotic agents.

Previously in preclinical studies on rabbits, it was shown that maximum concentration of Longidaze in blood serum after intramuscular injection is reached after 15 min (Appendix A). It was found that ^3^H-Longidaze after intravenous and intramuscular administration to rabbits is well distributed in the body, and the steady-state volume of distribution was 3.43 and 3.58 L/kg, respectively. The semi-elimination period of 3H-Longidaze after intravenous and intramuscular administration was 32.6 and 32.7 h, respectively. Bioavailability of Longidaze was about 96% after intramuscular administration. In the study of metabolism, it was found that Longidaze at a concentration up to 10 µM does not inhibit the isoenzymes 1A2, 2C9, 2C19, 2D6, 2C8, and 3A4 of human cytochrome P-450. The data obtained give grounds for prescribing Longidaze together with other drugs without fear of drug interaction.

In this study, the effect of various doses and routes of administration of Longidaze on the development of bleomycin-induced pulmonary fibrosis in mice was evaluated in comparison to the well-known drug pirfenidone. However, pirfenidone has a number of side effects in addition to its antifibrotic activity. Gastrointestinal and skin reactions are the most common adverse events caused by pirfenidone treatment [32]. In contrast to pirfenidone, administration of Longidaze at all doses tested did not adversely affect body weight in mice with pulmonary fibrosis. At the highest intramuscular dose of 1200 U/kg, the drug even contributed to an increase in body weight of mice with pulmonary fibrosis compared to untreated animals.

The early stage of bleomycin-induced lung injury is manifested by acute inflammation, including alveolar epithelial damage, inflammatory cell infiltration, release of inflammatory mediators, and peripheral blood leukocytosis [33]. It should be noted that during the acute phase of bleomycin-induced inflammation, macrophages and lymphocytes migrate to the lungs. The result of the inflammation secreted by the cells is seen in the increase of inflammatory cytokines in the lungs and leukocytes in the peripheral blood. It is known that levels of blood neutrophil and lymphocytes are associated with more rapid lung function decline in IPF [34]. Moreover, neutrophil count in circulation rises in systemic inflammation. On d7 of the study, Longidaze had anti-inflammatory activity; there was a decrease in the total number of leukocytes in the blood, the level of proinflammatory cytokines IL-6 and TNF-α, as well as the component of extracellular matrix HA, decreased in the lungs, and peribronchial and perivascular inflammation decreased compared to groups of mice with pulmonary fibrosis without treatment. It should be noted that the effect on the level of various leukocyte populations in the blood was dose-dependent after intramuscular administration of Longidaze, and no clear dose dependence was observed after intranasal administration. The reference drug pirfenidone also contributed to the reduction of leukocytosis in the peripheral blood of the animals, but had little effect on the severity of the inflammatory response in the lungs of the mice on d7 of the experiment.

Pulmonary inflammation is directly associated with proinflammatory cytokines such as IL-6 and TNF-α. Furthermore, a correlation between HA deposition and inflammatory cell, cytokine, and chemokine expression has been shown [35]. The accumulation of HA in the peribronchial and perivascular regions of the lung is an early event in the acute phase of lung inflammation [35]. In this study, we showed that treatment with Longidaze, which catalyzes the degradation of HA, reduced bleomycin-induced inflammatory response in the lung parenchyma. Summarizing the results obtained on d7 after bleomycin administration, it can be concluded that the highest anti-inflammatory activity of the drug Longidaze is recorded with intramuscular injection at the dose of 1200 U/kg, and the effect is superior to that of pirfenidone. At the same time, Longidaze does not have the side effects associated with pirfenidone. The anti-inflammatory effect of Longidaze at low and medium doses (both routes of administration) is comparable to that of the reference drug.

Importantly, uncontrolled lung injury is a hallmark of the initiation and progression of IPF, resulting in the release of proinflammatory and profibrotic cytokines, leading to further fibrosis-associated immune cell influx and ECM remodeling. Our data showed that Longidaze had anti-inflammatory and antifibrotic activity on d21 after bleomycin administration. It prevented the infiltration of inflammatory cells into peribronchial and perivascular spaces, reduced the expression of proinflammatory mediators (IL-6, TNF-α, hyaluronic acid) in lung tissue, and also reduced the activity of synthesis and deposition of connective tissue. The most pronounced positive effect on d21 after bleomycin administration was observed with Longidaze at a dose of 1200 U/kg intramuscularly and 120 U/kg intranasally.

A number of cytokines have been shown to stimulate fibrotic events, including TGF-β, TNF-α, and IL-6. The best studied of these various cytokines in pulmonary fibrosis is TGF-β. TGF-β is recognized as an important regulator of tissue fibrosis in general, and numerous studies have convincingly confirmed its role in pulmonary fibrosis in particular [36]. It has been shown that TGF-β is required for the development of bleomycin-induced pulmonary fibrosis, which is a potent stimulator of HA production, and promotes the transition of resident fibroblasts to myofibroblasts in vitro [37]. The antifibrotic effect of Longidaze in the pulmonary fibrosis model was manifested by a decrease in the area of connective tissue in the lungs of mice and a decrease in the levels of profibrotic mediators (TGF-β1, type 1 collagen, hydroxyproline) in the lung tissue. At the same time, the effect of intramuscular application of Longidaze at doses of 120 U/kg and 1200 U/kg is most pronounced, and exceeds that of the reference drug pirfenidone. Thus, the investigated drug Longidaze showed significant anti-inflammatory and antifibrotic activity in the bleomycin-induced model of pulmonary fibrosis. The highest therapeutic efficacy is observed with the use of Longidaze at doses of 120 and 1200 U/kg intramuscularly, which is superior to that of the reference drug pirfenidone. It should be noted that Longidaze does not have the side effects associated with pirfenidone.

However, our study has some limitations. We performed a preliminary study to determine the active dose of Longidaze and the best route of administration in bleomycin-induced pulmonary fibrosis. It is not possible to study many of the important mechanisms of drug action in a single study. To assess the mechanism of Longidaze, the results presented here are required further validations using immunohistochemistry and cytometry studies, etc. It is important to determine which cells are affected by Longidaze.

## 5. Conclusions

In conditions of modeling of bleomycin damage of alveolar epithelium, Longidaze impedes the synthesis and deposition of collagen in the lungs of mice. Moreover, Longidaze affects the inflammatory mediators. The data presented in this study indicate that Longidaze is a new promising drug for the treatment of IPF.

## Figures and Tables

**Figure 1 life-13-01932-f001:**
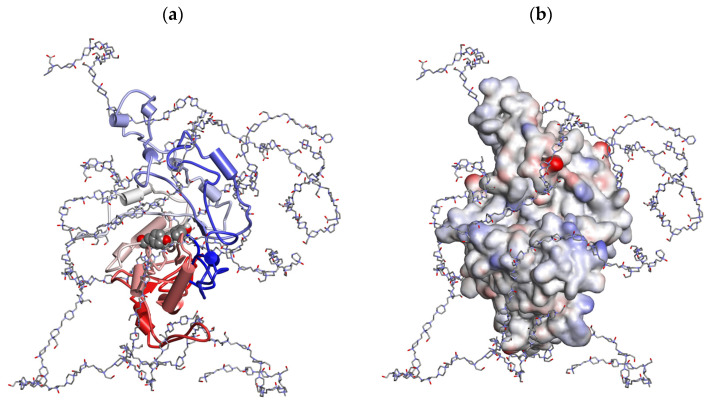
Active site view of the in silico visualized hyaluronidase conjugate. Two active site tyrosine residues are shown in CPK format with N- to C-terminal (blue to red color) protein chain visualization (**a**). A soft interpolated charge surface of the protein is added on the right (**b**). Only for visualization purposes of possible conformational outlook of conjugated hyaluronidase molecule (e.g., not for accurate calculation or prediction of drug structure), all in silico experiments were performed using the bovine Hyal1 protein sequence and homology modeling algorithm, manual attachment of the random polymer carboxyl groups to random lysine residues located on the protein surface, followed by standard minimization cascade algorithm and force field minimization using the molecular mechanics algorithm CHARMm (as implemented in Discovery Studio 2.5, Accelrys, San Diego, CA, USA). The crystal structure of human PH-20 (2PE4.pdb) with hydrogen atoms added was used as the starting template structure.

**Figure 2 life-13-01932-f002:**
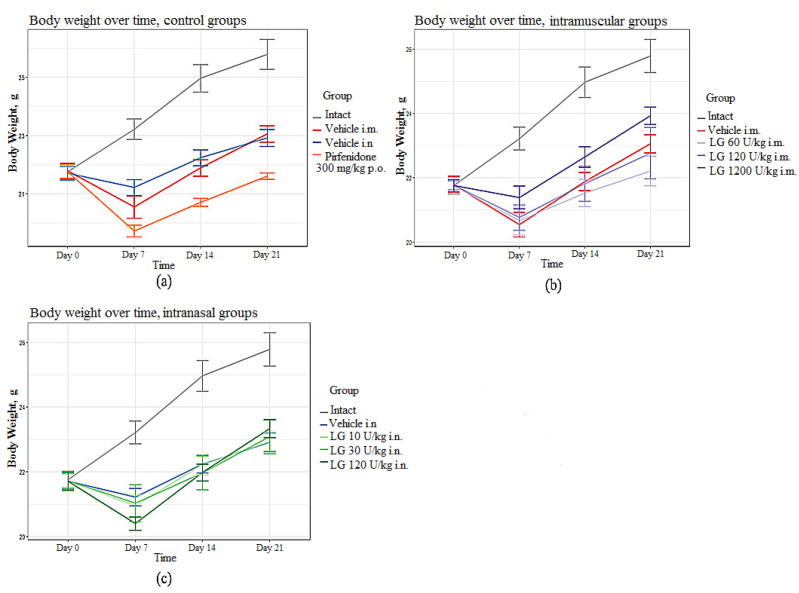
Body weight in C57BL/6 mice from group of intact control, from group with pulmonary fibrosis after pirfenidone administration per os (**a**), after intramuscular (**b**) and intranasal (**c**) administration of various doses of Longidaze compared to mice without treatment. P.o.—per os, i.m.—intramuscular, i.n.—intranasal.

**Figure 3 life-13-01932-f003:**
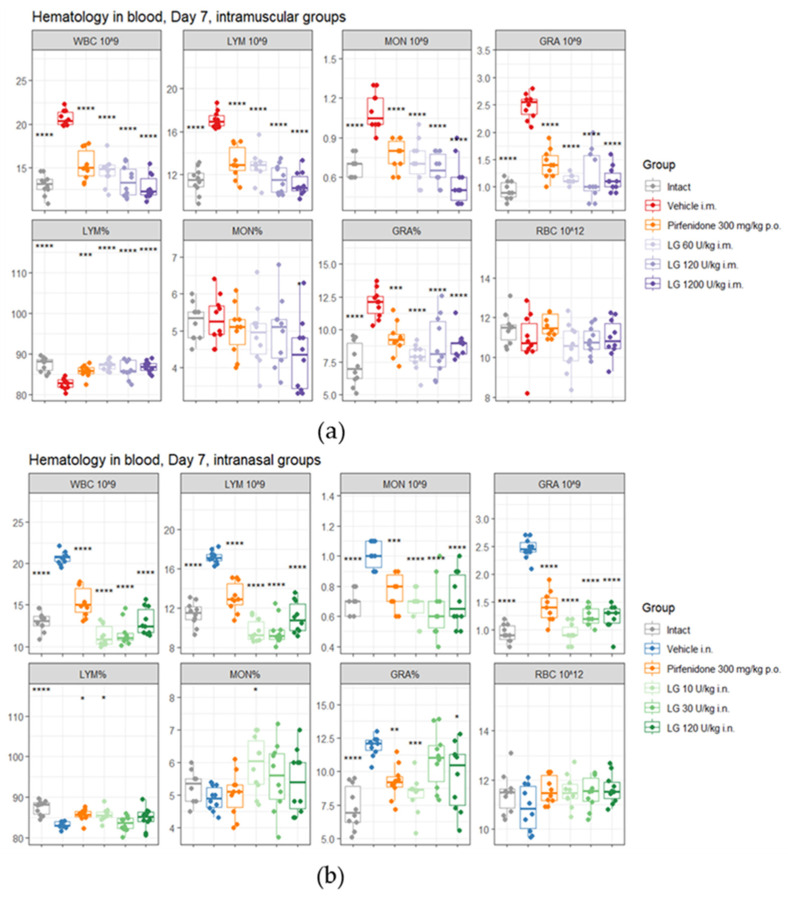
The effect of Longidaze on the hematological parameters of C57BL/6 mice modeling bleomycin-induced pulmonary fibrosis on d7. Hematological parameters after pirfenidone administration per os, after intramuscular (**a**) and intranasal (**b**) administration of different doses of Longidaze. Groups of mice: Intact—mice of intact control (group 1); Vehicle i.m. and Vehicle i.n.—mice treated with bleomycin (groups 2 and 3, respectively); pirfenidone 300 mg/kg p.o.—pulmonary fibrosis + pirfenidone (group 4); LG 60 U/kg i.m.—pulmonary fibrosis + Longidaze intramuscularly at a dose of 60 U/kg (group 5); LG 120 U/kg i.m.—pulmonary fibrosis + Longidaze intramuscularly at a dose of 120 U/kg (group 6); LG 1200 U/kg i.m.—pulmonary fibrosis + Longidaze intramuscularly at a dose of 1200 U/kg (group 7); LG 10 U/kg i.n.—pulmonary fibrosis + Longidaze intranasally at a dose of 10 U/kg (group 8); LG 30 U/kg i.n.—pulmonary fibrosis + Longidaze intranasal at a dose of 30 U/kg (group 9); LG 120 U/kg i.n.—pulmonary fibrosis + Longidaze intranasal at a dose of 120 U/kg (group 10). P.o.—per os, i.m.—intramuscular, i.n.—intranasal. WBCs—white blood cells (10^9^/L blood); GRA 10^9^, LYM 10^9^, MON 10^9^ (absolute number), GRA %, LYM %, MON % (relative number)—number of neutrophils, lymphocytes, monocytes; RBCs 10^12^—red blood cells (10^12^/L blood). Statistical analysis of data was performed using the Kruskal–Wallis test, post hoc Wilcoxon test for pairwise comparisons with bleomycin control group (with corresponding route of administration), and Bonferroni correction (ns—*p* > 0.05, * *p* < 0.05, ** *p* < 0.01, *** *p* < 0.001, **** *p* < 0.0001).

**Figure 4 life-13-01932-f004:**
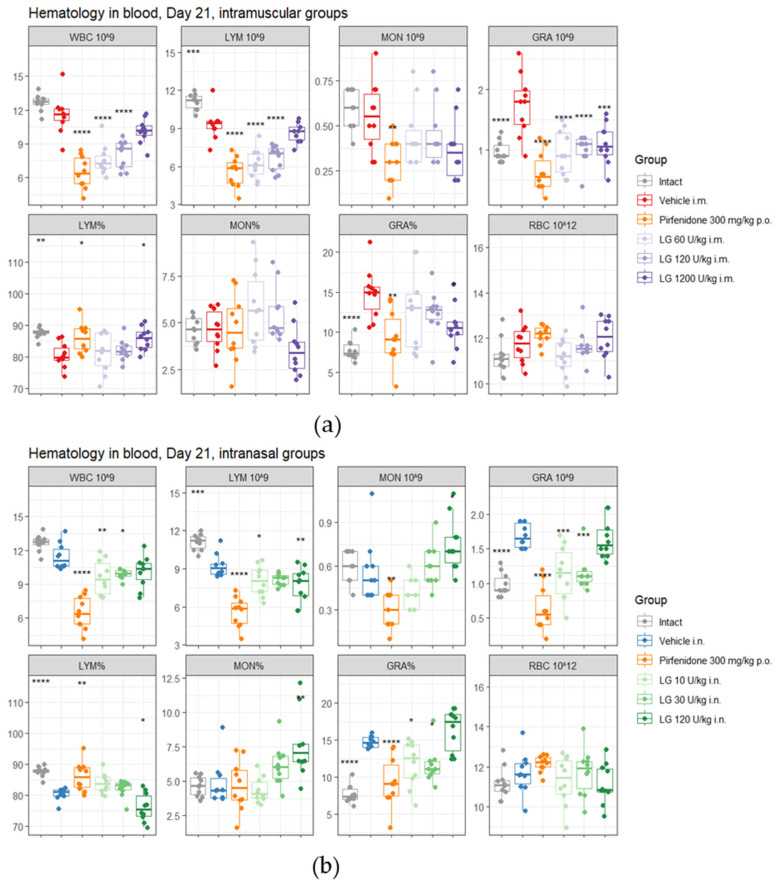
The effect of Longidaze on the hematological parameters of C57BL/6 mice modeling bleomycin-induced pulmonary fibrosis on d21. Hematological parameters after pirfenidone administration per os, after intramuscular (**a**) and intranasal (**b**) administration of different doses of Longidaze. Groups of mice: Intact—mice of intact control (group 1); Vehicle i.m. and Vehicle i.n.—mice treated with bleomycin (groups 2 and 3, respectively); pirfenidone 300 mg/kg p.o.—pulmonary fibrosis + pirfenidone (group 4); LG 60 U/kg i.m.—pulmonary fibrosis + Longidaze intramuscularly at a dose of 60 U/kg (group 5); LG 120 U/kg i.m.—pulmonary fibrosis + Longidaze intramuscularly at a dose of 120 U/kg (group 6); LG 1200 U/kg i.m.—pulmonary fibrosis + Longidaze intramuscularly at a dose of 1200 U/kg (group 7); LG 10 U/kg i.n.—pulmonary fibrosis + Longidaze intranasally at a dose of 10 U/kg (group 8); LG 30 U/kg i.n.—pulmonary fibrosis + Longidaze intranasal at a dose of 30 U/kg (group 9); LG 120 U/kg i.n.—pulmonary fibrosis + Longidaze intranasal at a dose of 120 U/kg (group 10). P.o.—per os, i.m.—intramuscular, i.n.—intranasal. WBCs—white blood cells (10^9^/L blood); GRA 10^9^, LYM 10^9^, MON 10^9^ (absolute number), GRA %, LYM %, MON % (relative number)—number of neutrophils, lymphocytes, monocytes; RBCs 10^12^—red blood cells (10^12^/L blood). Statistical analysis of data was performed using the Kruskal–Wallis test, post hoc Wilcoxon test for pairwise comparisons with bleomycin control group (with corresponding route of administration), and Bonferroni correction (ns—*p* > 0.05, * *p* < 0.05, ** *p* < 0.01, *** *p* < 0.001, **** *p* < 0.0001).

**Figure 5 life-13-01932-f005:**
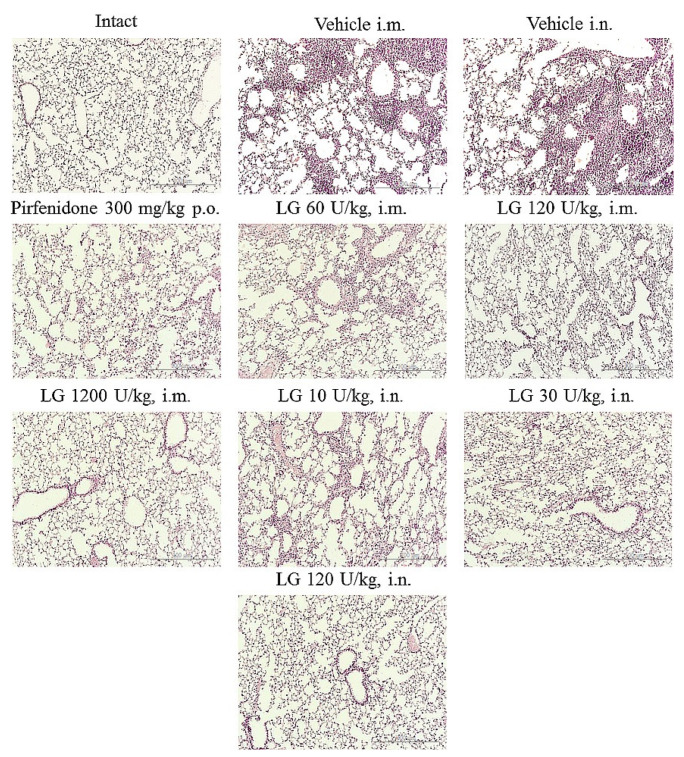
Histological analysis of lung inflammation on d7. Representative photomicrographs of lung tissue sections obtained from male C57BL/6 mice and stained with H&E. ×100. Scale bar 50 μm. Groups: Intact—mice of intact control (group 1); Vehicle i.m. and Vehicle i.n.—mice treated with bleomycin (groups 2 and 3, respectively); pirfenidone 300 mg/kg p.o.—pulmonary fibrosis + pirfenidone (group 4); LG 60 U/kg i.m.—pulmonary fibrosis + Longidaze intramuscularly at a dose of 60 U/kg (group 5); LG 120 U/kg i.m.—pulmonary fibrosis + Longidaze intramuscularly at a dose of 120 U/kg (group 6); LG 1200 U/kg i.m.—pulmonary fibrosis + Longidaze intramuscularly at a dose of 1200 U/kg (group 7); LG 10 U/kg i.n.—pulmonary fibrosis + Longidaze intranasally at a dose of 10 U/kg (group 8); LG 30 U/kg i.n.—pulmonary fibrosis + Longidaze intranasal at a dose of 30 U/kg (group 9); LG 120 U/kg i.n.—pulmonary fibrosis + Longidaze intranasal at a dose of 120 U/kg (group 10). P.o.—per os, i.m.—intramuscular, i.n.—intranasal.

**Figure 6 life-13-01932-f006:**
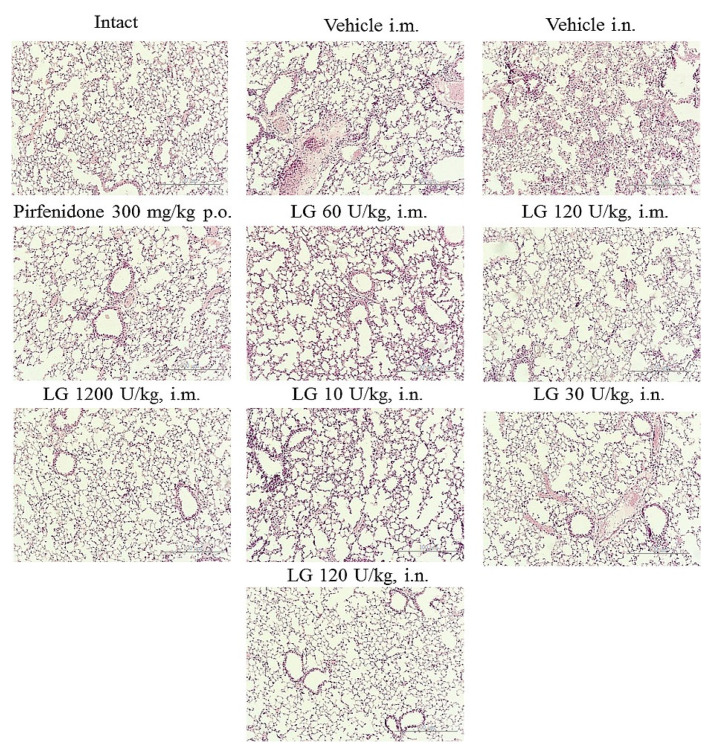
Photomicrographs of lung sections obtained from male C57BL/6 mice on d21. Tissues were stained with H&E. ×100. Scale bar 50 μm. Groups: Intact—intact control mice (group 1); Vehicle i.m. and Vehicle i.n.—bleomycin treated mice (groups 2 and 3, respectively); pirfenidone 300 mg/kg p.o.—pulmonary fibrosis + pirfenidone (group 4); LG 60 U/kg i.m.—pulmonary fibrosis + Longidaze intramuscularly at a dose of 60 U/kg (group 5); LG 120 U/kg i.m.—pulmonary fibrosis + Longidaze intramuscularly at a dose of 120 U/kg (group 6); LG 1200 U/kg i.m.—pulmonary fibrosis + Longidaze intramuscularly at a dose of 1200 U/kg (group 7); LG 10 U/kg i.n.—pulmonary fibrosis + Longidaze intranasally at a dose of 10 U/kg (group 8); LG 30 U/kg i.n.—pulmonary fibrosis + Longidaze intranasal at a dose of 30 U/kg (group 9); LG 120 U/kg i.n.—pulmonary fibrosis + Longidaze intranasal at a dose of 120 U/kg (group 10). P.o.—per os, i.m.—intramuscular, i.n.—intranasal.

**Figure 7 life-13-01932-f007:**
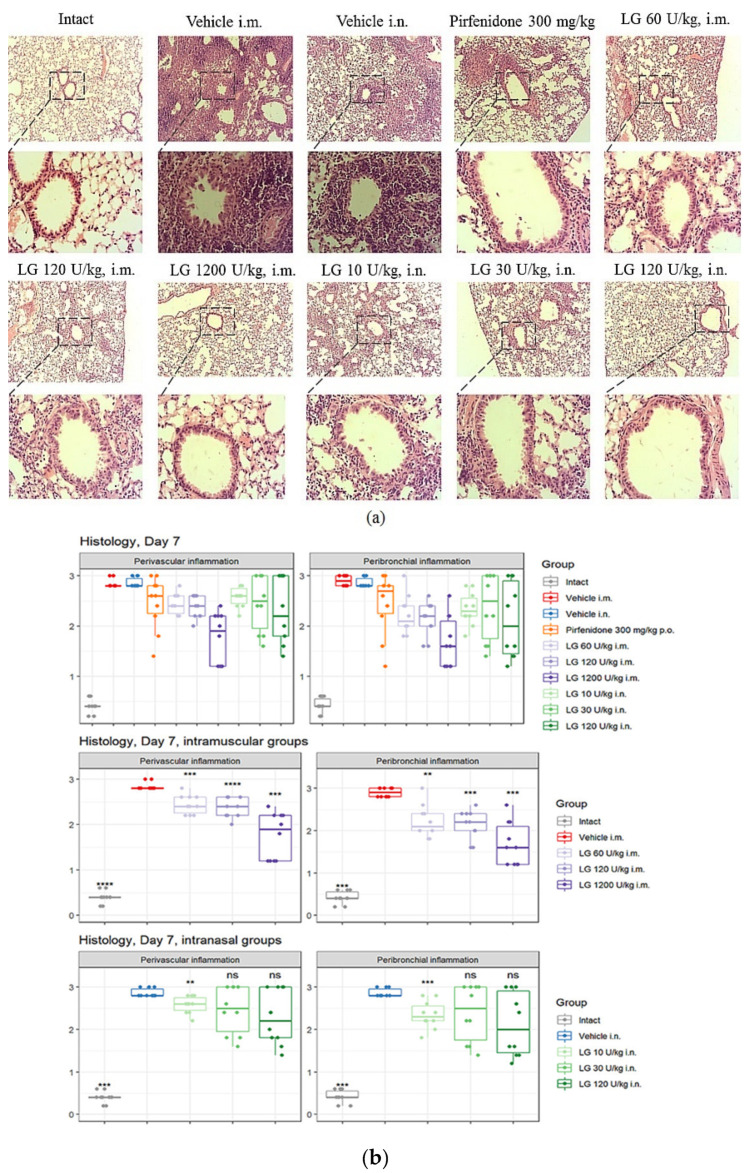
Effect of Longidaze on perivascular and peribronchial inflammation in the lungs of C57BL/6 mice on d7. Representative images of lung tissue section stained with H&E (**a**) and its respective inflammation scores (**b**). Groups: Intact—mice of intact control (group 1); Vehicle i.m. and Vehicle i.n.—mice treated with bleomycin (groups 2 and 3, respectively); pirfenidone 300 mg/kg p.o.—pulmonary fibrosis + pirfenidone (group 4); LG 60 U/kg i.m.—pulmonary fibrosis + Longidaze intramuscularly at a dose of 60 U/kg (group 5); LG 120 U/kg i.m.—pulmonary fibrosis + Longidaze intramuscularly at a dose of 120 U/kg (group 6); LG 1200 U/kg i.m.—pulmonary fibrosis + Longidaze intramuscularly at a dose of 1200 U/kg (group 7); LG 10 U/kg i.n.—pulmonary fibrosis + Longidaze intranasally at a dose of 10 U/kg (group 8); LG 30 U/kg i.n.—pulmonary fibrosis + Longidaze intranasal at a dose of 30 U/kg (group 9); LG 120 U/kg i.n.—pulmonary fibrosis + Longidaze intranasal at a dose of 120 U/kg (group 10). P.o.—per os, i.m.—intramuscular, i.n.—intranasal. Statistical analysis of data was performed using the Kruskal–Wallis test, post hoc Wilcoxon test for pairwise comparisons with the bleomycin control group (with appropriate route of administration), and Bonferroni correction (ns—*p* > 0.05, ***p* < 0.01, *** *p* < 0.001, **** *p* < 0.0001).

**Figure 8 life-13-01932-f008:**
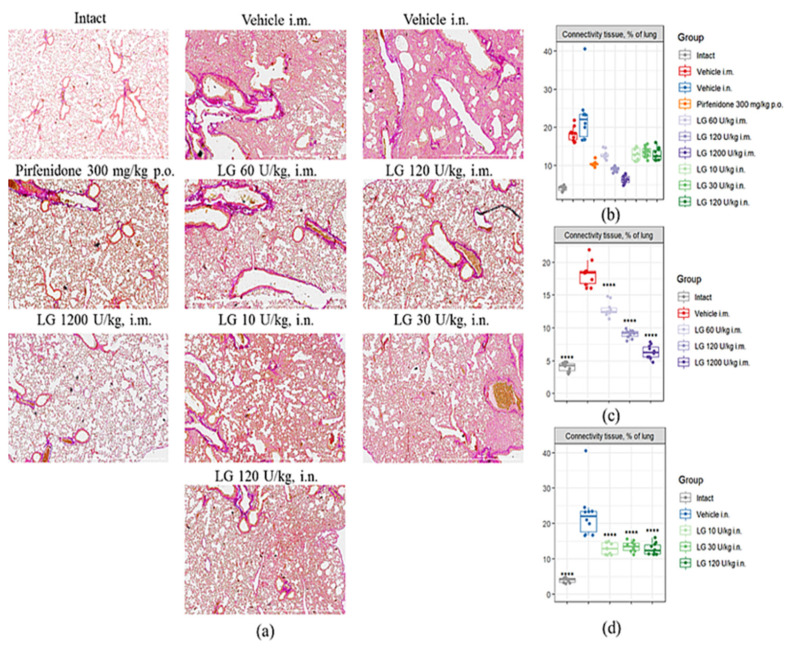
Effects of Longidaze on the content of connective tissue in the lungs of C57BL/6 mice on d21. (**a**) Photomicrographs of left lung sections (middle pulmonary field) obtained from male C57BL/6 mice on d21. Tissues stained by Van Gieson, scale bar 100 μm. (**b**) Content of the connective tissue in the lungs of C57BL/6 mice from all groups; (**c**) content of the connective tissue in the lungs of C57BL/6 mice from groups after intramuscularly administration; (**d**) content of the connective tissue in the lungs of C57BL/6 mice from groups after intranasal administration. Groups: Intact—mice of intact control (group 1); Vehicle i.m. and Vehicle i.n.—mice treated with bleomycin (groups 2 and 3, respectively); pirfenidone 300 mg/kg p.o.—pulmonary fibrosis + pirfenidone (group 4); LG 60 U/kg i.m.—pulmonary fibrosis + Longidaze intramuscularly at a dose of 60 U/kg (group 5); LG 120 U/kg i.m.—pulmonary fibrosis + Longidaze intramuscularly at a dose of 120 U/kg (group 6); LG 1200 U/kg i.m.—pulmonary fibrosis + Longidaze intramuscularly at a dose of 1200 U/kg (group 7); LG 10 U/kg i.n.—pulmonary fibrosis + Longidaze intranasally at a dose of 10 U/kg (group 8); LG 30 U/kg i.n.—pulmonary fibrosis + Longidaze intranasal at a dose of 30 U/kg (group 9); LG 120 U/kg i.n.—pulmonary fibrosis + Longidaze intranasal at a dose of 120 U/kg (group 10). P.o.—per os, i.m.—intramuscular, i.n.—intranasal. Statistical analysis of data was performed using the Kruskal–Wallis test, post hoc Wilcoxon test for pairwise comparisons with the bleomycin control group (with appropriate route of administration), and Bonferroni correction (ns—*p* > 0.05, **** *p* < 0.0001).

**Figure 9 life-13-01932-f009:**
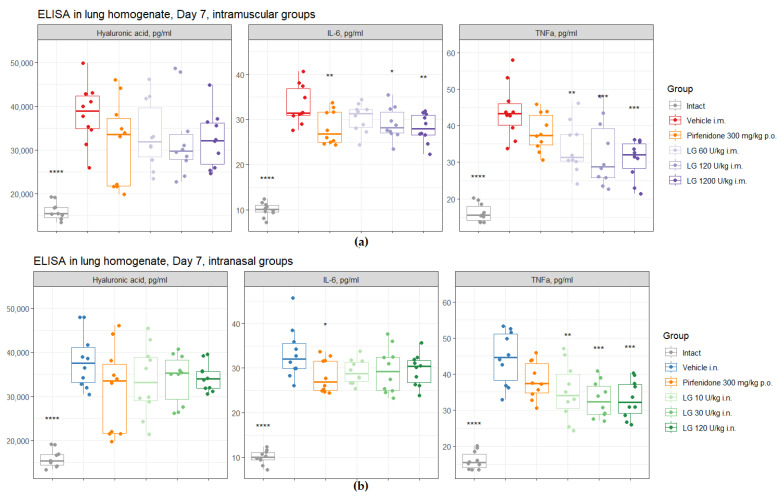
Effects of Longidaze treatment on the IL-6, TNF-α, and hyaluronic acid levels in homogenate of right lung lobes received from male C57BL/6 mice on d7. ELISA parameters after pirfenidone administration per os, after intramuscular (**a**) and intranasal (**b**) administration of different doses of Longidaze. Groups of mice: Intact—mice of intact control (group 1); Vehicle i.m. and Vehicle i.n.—mice treated with bleomycin (groups 2 and 3, respectively); pirfenidone 300 mg/kg p.o.—pulmonary fibrosis + pirfenidone (group 4); LG 60 U/kg i.m.—pulmonary fibrosis + Longidaze intramuscularly at a dose of 60 U/kg (group 5); LG 120 U/kg i.m.—pulmonary fibrosis + Longidaze intramuscularly at a dose of 120 U/kg (group 6); LG 1200 U/kg i.m.—pulmonary fibrosis + Longidaze intramuscularly at a dose of 1200 U/kg (group 7); LG 10 U/kg i.n.—pulmonary fibrosis + Longidaze intranasally at a dose of 10 U/kg (group 8); LG 30 U/kg i.n.—pulmonary fibrosis + Longidaze intranasal at a dose of 30 U/kg (group 9); LG 120 U/kg i.n.—pulmonary fibrosis + Longidaze intranasal at a dose of 120 U/kg (group 10). P.o.—per os, i.m.—intramuscular, i.n.—intranasal. Statistical analysis of data was performed using the Kruskal–Wallis test, post hoc Wilcoxon test for pairwise comparisons with the bleomycin control group (with appropriate route of administration), and Bonferroni correction (ns—*p* > 0.05, * *p* < 0.05, ** *p* < 0.01, *** *p* < 0.001, **** *p* < 0.0001).

**Figure 10 life-13-01932-f010:**
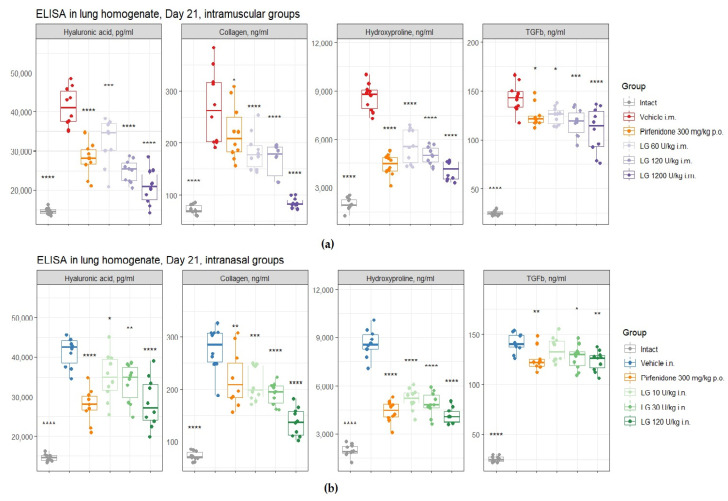
Effects of Longidaze treatment on hyaluronic acid, collagen I, hydroxyproline, and TGF-β1 levels in right lung lobe homogenate obtained from male C57BL/6 mice on d21. ELISA parameters after pirfenidone administration per os, after intramuscular (**a**) and intranasal (**b**) administration of different doses of Longidaze. Groups of mice: Intact—mice of intact control (group 1); Vehicle i.m. and Vehicle i.n.—mice treated with bleomycin (groups 2 and 3, respectively); pirfenidone 300 mg/kg p.o.—pulmonary fibrosis + pirfenidone (group 4); LG 60 U/kg i.m.—pulmonary fibrosis + Longidaze intramuscularly at a dose of 60 U/kg (group 5); LG 120 U/kg i.m.—pulmonary fibrosis + Longidaze intramuscularly at a dose of 120 U/kg (group 6); LG 1200 U/kg i.m.—pulmonary fibrosis + Longidaze intramuscularly at a dose of 1200 U/kg (group 7); LG 10 U/kg i.n.—pulmonary fibrosis + Longidaze intranasally at a dose of 10 U/kg (group 8); LG 30 U/kg i.n.—pulmonary fibrosis + Longidaze intranasal at a dose of 30 U/kg (group 9); LG 120 U/kg i.n.—pulmonary fibrosis + Longidaze intranasal at a dose of 120 U/kg (group 10). P.o.—per os, i.m.—intramuscular, i.n.—intranasal. Statistical analysis of data was performed using the Kruskal–Wallis test, post hoc Wilcoxon test for pairwise comparisons with the bleomycin control group (with appropriate route of administration), and Bonferroni correction (ns—*p* > 0.05, * *p* < 0.05, ** *p* < 0.01, *** *p* < 0.001, **** *p* < 0.0001).

## Data Availability

Not applicable.

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
