# Peer review of "Anti-Inflammatory and Antifibrotic Potential of Longidaze in Bleomycin-Induced Pulmonary Fibrosis"

_life, 2023, doi:10.3390/life13091932_

Round 1
Reviewer 1 Report
life-2538054
The authors address a very interesting topic: Idiopathic pulmonary fibrosis (IPF), a lung disease characterized by progressive parenchymal fibrosis and respiratory failure. In a model of bleomycin-induced pulmonary fibrosis, the antifibrotic and anti-inflammatory activity of Longidaza (bovine hyaluronidase azoximer), which contains a conjugate of the hyaluronidase enzyme with a synthetic high molecular weight azoximer bromide carrier, was investigated in C57BL/6 mice. Innovative: Longidaza reduces lung fibrosis as evidenced by improvement in histopathological lung damage, reduction in connective tissue area and levels of profibrotic factors (TGF-β1, hydroxyproline, collagen I) in lung tissue. In addition, Longidaza inhibited the inflammatory response in pulmonary fibrosis, reducing the levels of IL-6, TNF-α, hyaluronic acid in the lung tissue and the recruitment of inflammatory cells in the lung tissue. The highest therapeutic efficacy was observed with the use of Longidaza in doses of 120 and 1200 U/kg intramuscularly, which was better than that of the reference drug pirfenidone axunio.
Recommendations: 1. Moderation of the English language. 2. In the introduction, it is good to present the characteristics of the used protector. 3. It would be good to include antioxidant enzyme levels to monitor the effects in the body. 4. Figure 2 have to be divided - for greater clarity. 5. The discussion is insufficient, to be supplemented.
Minor editing of English language required
Author Response
We thank the reviewers for their time and valuable comments. We have now revised the manuscript according to the suggestions. Below are the reviewers' comments and our responses. All changes are included in the revised manuscript.
Recommendations:
- In the introduction, it is good to present the characteristics of the used protector.
We have added information to the introduction.
- It would be good to include antioxidant enzyme levels to monitor the effects in the body.
Thank you for your question. We agree that part of the altered alveolar environment in pulmonary fibrosis involves oxidative stress driven by an imbalance between oxidant production and antioxidant defenses.
The aim of our study was to investigate the effect of Longidaza on the development of bleomycin-induced lung fibrosis and to compare the effects of Longidaza with those of pirfenidone. In addition, Longidaza was administered at different doses and by different routes. This was a screening study. Unfortunately, we were not able to explore many of the important mechanisms of action of drugs in a single study. It is difficult to present all the results in a single article. We plan to continue our study.
- Figure 2 have to be divided - for greater clarity.
We have now split the Figure 2 into two parts
- The discussion is insufficient, to be supplemented.
We have now added information to the discussion.
Reviewer 2 Report
There are some concerns. The paper has potential. There is a clear conflict of interest with Petrovax that is not declared. The big concern has to do with "inflammatory score". Primary data indicating what was detected needs to be shown. These days, H and E are insufficient. IHC with specific markers is necessary.
1) no external funding source is listed. However, several authors are from a company, Petrovax. Please list specifically the source of funding, external or internal, and please include a separate, extremely detailed and specific conflict of interest statement that mentions that 2 authors are employees of a company that has IP rights concerning the drug evaluated, a funding statement that the company supported the work, whether the company had a say regarding the decision to publish, as appropriate.
2 what is the relationship, if any, between poloxamergaluronidase and Longidaza. Please supply more details outlining specifically what Longidaza is, how this compound was identified, with references. It appears that some of this information is in "Methods". Relevant details need to be included in Introduction.
3. Fig 2. Post-hoc test pairwise with "bleomycin control" is used. This may be OK in a sense, but the authors are also also doing a dose response and mode of administration. As multiple variables are being compared, an appropriate test needs to be conducted. I would like to have the dose response and the mode of administration also evaluated. Significance needs to be indicated on the graph itself. The abbreviations im, in and po need to be defined
4. Figure 3 and 4 The abbreviations im and in need to be defined. Scale bar needs to be included. That H and E sections were examined needs to be stated in the title, not the last sentence.
5. Both Figures 3 and 4 seem to have identical information. What is the purpose of each figure? What are the conclusions? Please explain in the title and legend. Figure 3 and 4 do not seem to be quantified. Are these representative images? N=? What details are we supposed to look at? Show with arrows, and explain in the legend and title
6. Figure 5 shows a histogram derived from primary data. The primary data needs to be shown on the figure. What inflammatory cells were detected? Macrophage, T cells etc? Show immunohistochemistry detecting individual cell types. Pairwise with bleomycin is a slightly misleading posthoc test, as multiple groups (dose response) is being used. Is there a dose response?
7. Figure 6. What is "connectivity tissue". All letters need to be upper left of panel, capitals, no parentheses. I do not understand the purpose of 6B. Include perfinidone in 6C, D. Pairwise with bleomycin is a slightly misleading posthoc test, as multiple groups (dose response) is being used. Is there a dose response?\
8. Figure 7. See previous comments regarding statistics. Significance is indicated nowhere on the graph.
9. Figure 8. Text and figure legends. How was each marker assessed? ELISA? See previous comments regarding statistics. Significance is indicated nowhere on the graph. Indicate "TGFbeta1" on graph
10. I am not sure that the authors can comment on side effects of the drug, as I do not see any appropriate assays in this current paper
Author Response
We thank the reviewers for their time and valuable comments. We have now revised the manuscript according to the suggestions. Below are the reviewers' comments and our responses. All changes have been incorporated in the revised version.
- no external funding source is listed. However, several authors are from a company, Petrovax. Please list specifically the source of funding, external or internal, and please include a separate, extremely detailed and specific conflict of interest statement that mentions that 2 authors are employees of a company that has IP rights concerning the drug evaluated, a funding statement that the company supported the work, whether the company had a say regarding the decision to publish, as appropriate.
Response.
In the Funding and Acknowledgements we have now added the following information.
2) what is the relationship, if any, between poloxamergaluronidase and Longidaza. Please supply more details outlining specifically what Longidaza is, how this compound was identified, with references. It appears that some of this information is in "Methods". Relevant details need to be included in Introduction.
Response.
In our publication, the Longidaza is a new formulation of hyaluronidase. We studied the pharmacological activity of a new drug. We did not study and compare the effect of poloxamer hyaluronidase and Longidaza in animals with bleomycin-induced fibrosis. These are different hyaluronidase-based drugs. We have now added information in the Introduction and Methods.
3) Fig 2. Post-hoc test pairwise with "bleomycin control" is used. This may be OK in a sense, but the authors are also also doing a dose response and mode of administration. As multiple variables are being compared, an appropriate test needs to be conducted. I would like to have the dose response and the mode of administration also evaluated. Significance needs to be indicated on the graph itself. The abbreviations im, in and po need to be defined
Response.
We now have divided Figure 2 and added the information in Figure legend.
4) Figure 3 and 4 The abbreviations im and in need to be defined. Scale bar needs to be included. That H and E sections were examined needs to be stated in the title, not the last sentence.
Response.
We have now corrected it. We have added information to the legends of the figures.
5) Both Figures 3 and 4 seem to have identical information. What is the purpose of each figure? What are the conclusions? Please explain in the title and legend. Figure 3 and 4 do not seem to be quantified. Are these representative images? N=? What details are we supposed to look at? Show with arrows, and explain in the legend and title
Response.
We have now corrected it. We have now added information to the figures. Figure 5 (was Figure 3) shows photomicrographs of lung where tissue was stained with hematoxylin-eosin at d7 of the experiment. Figure 6 (was Figure 4) shows photomicrographs of lung where tissue was stained with hematoxylin-eosin at d21 of the experiment.
6) Figure 5 shows a histogram derived from primary data. The primary data needs to be shown on the figure. What inflammatory cells were detected? Macrophage, T cells etc? Show immunohistochemistry detecting individual cell types. Pairwise with bleomycin is a slightly misleading posthoc test, as multiple groups (dose response) is being used. Is there a dose response?
Response.
Thank you for your question.
Hematoxylin and eosin (H&E) staining is a common technique for staining pathological tissues, including assessment of lung architecture and inflammatory cells. In our investigation, we used this method to assess lung architecture and inflammatory cells. We are aware that our research has limitations. We observed mononuclear infiltration of lung tissue in bleomycin-induced pulmonary fibrosis. We detected macrophages, T cells, and other individual cell types. Unfortunately, we did not perform separate cell counts. In addition, we evaluated the levels of pro-inflammatory mediators (IL-6, TGF-β, TNF-α), hyaluronic acid, hydroxyproline, and collagen type I. This was a preliminary study to determine the active dose of Longidaza. Next, we plan to investigate the mechanism of action of Longidaza and conduct an immunohistochemistry study.
7) Figure 6. What is "connectivity tissue". All letters need to be upper left of panel, capitals, no parentheses. I do not understand the purpose of 6B. Include perfinidone in 6C, D. Pairwise with bleomycin is a slightly misleading posthoc test, as multiple groups (dose response) is being used. Is there a dose response?
Response.
We now have a corrected image. To visualize collagen deposition, we used the Van Gieson method of picrofuchsin staining. In addition, we evaluated collagen type 1 and hydroxyproline by ELISA assay. Such an approach to the study of connective tissue in the lungs is used by many authors [1-6]. We will certainly consider all your comments and suggestions in our further investigations.
- Prentø P. Van Gieson's picrofuchsin. The staining mechanisms for collagen and cytoplasm, and an investigation of the dye diffusion rate model of differential staining. Histochemistry. 1993 Feb;99(2):163-74.
- Singh M, Chaudhary AK, Pandya S, Debnath S, Singh M, Singh PA, Mehrotra R. Morphometric analysis in potentially malignant head and neck lesions: oral submucosal fibrosis. Asian Pac J Cancer Prev. 2010;11(1):257-60.
3 Fukuda C, Goto K, Imamura M, Nakamura T. Bioactive bone cement with low titania particle content without postsilanization: effect of filler content on osteoconductivity, mechanical properties, and handling characteristics J Biomed Mater Res B Appl Biomater. 2010 Nov;95(2):407-13. doi: 10.1002/jbm.b.31731.
- Zhan YT, Li L, Weng J, Song X, Yang SQ, An W. Serum autofluorescence, a potential serum marker for the diagnosis of liver fibrosis in rats. Int J Mol Sci. 2012;13(9):12130-9. doi: 10.3390/ijms130912130.
- Hunasgi S, Koneru A, Vanishree M, Shamala R. Keloid: A case report and review of the pathophysiology and differences between keloid and hypertrophic scars. J Oral Maxillofac Pathol. 2013 Jan;17(1):116-20. doi: 10.4103/0973-029X.110701.
- Nasrollahi Nia, F., Asadi, A., Zahri, S., & Abdolmaleki, A. (2020). Biosynthesis, characterization and evaluation of supporting properties and biocompatibility of DBM nanoparticles on a tissue-engineered nerve conduit from decellularized sciatic nerve. Regenerative Therapy, 14, 315-321.
8) Figure 7. See previous comments regarding statistics. Significance is indicated nowhere on the graph.
Response.
We have now corrected it. We have now added information to the images.
9) Figure 8. Text and figure legends. How was each marker assessed? ELISA? See previous comments regarding statistics. Significance is indicated nowhere on the graph. Indicate "TGFbeta1" on graph
Response.
We have now corrected it. We have now added information to the images.
10) I am not sure that the authors can comment on side effects of the drug, as I do not see any appropriate assays in this current paper
Response.
Thank you for your question. The aim of our study was to investigate the effect of Longidaza on the development of bleomycin-induced pulmonary fibrosis and to compare the effect of Longidaza with the effect of pirfenidone. We studied Longidaza at different doses and different routes of administration. This was a screening study. We did not observe any side effects of the drug.
Unfortunately, in a single study we could not explore many important aspects of the action of drugs. It is difficult to present all the results in a single article. We plan to continue our study.
Round 2
Reviewer 2 Report
The responses are disappointing.
The authors were non-responsive concerning the COI. No funding source (internal or external) has been included, nor has the position of the supplier, or owner, of the drug Petrovax been explained.
The authors were also non responsive regarding the use of statistical tests. A dose response was evaluated, but not included in the statistical analysis. Therefore the wrong statistical test has been used, as two variables (dose response and effect) are tested
I am fully aware that H and E sections were used historically to evaluate inflammation. Unfortunately, this is no longer considered adequate or precise. The authors have not provided any explanation regarding how they used H and E to look for inflammation, nor are any insets used to indicate what they are focusing on. The authors simply must use unbiased histological methods using specific antibodies.
Author Response
We are very grateful to the reviewer for his work in reviewing our manuscript.
The authors were non-responsive concerning the COI. No funding source (internal or external) has been included, nor has the position of the supplier, or owner, of the drug Petrovax been explained.
Response:
We thank the reviewers for their time and valuable comments. We have now revised the manuscript according to the suggestions. Below are the reviewers' comments and our responses. All changes have been incorporated in the revised version.
New formulation added to the publication.
Petrovax Pharma LLC provided Longidaza for our study. Petrovax employees helped us in writing—review and editing of manuscript, decor of figures and project administration. The Petrovax employees had no role in the design of the study, in the collection, analysis, or interpretation of the data.
Petrovax is a full-cycle biotechnology company, a leading Russian developer and manufacturer of original medicines and vaccines. The modern research center is in Moscow, RF. Petrovax holds patents in Russia and abroad for molecules and production technologies. Main office, Russia, 123112, Moscow, Presnenskaya Embankment, Vostok Federation Tower 12, 38th floor, Tel: +7 495 730-75-45 <info@petrovax.ru>
The authors declare that the research was conducted in the absence of any commercial or financial relationships that could be construed as a potential conflict of interest.
The authors were also non responsive regarding the use of statistical tests. A dose response was evaluated, but not included in the statistical analysis. Therefore the wrong statistical test has been used, as two variables (dose response and effect) are tested
Response:
We thank the reviewer for the question.
We have evaluated a dose response and performed the standard statistical analysis. Statistical significance was evaluated by Student’s t-test (for parametric data), or Mann–Whitney test (for nonparametric data) when appropriate. Statistical analysis of hematology and ELISA data was performed using the Kruskal-Wallis test, post-hoc Wilcoxon test for pairwise comparisons with the bleomycin control group (with appropriate route of administration), Bonferroni correction. These methods were used in scientific papers. It is generally assumed that there exists a well-defined relationship between drug dose and drug effect and that this can be expressed by a dose-response curve. However it was shown that there is no such clear relation and that the dose-response curve provides only limited information about the drug effect. We aimed to assess the efficacy of Longidaza compare to pirfenidone at the bleomycin-induced pulmonary fibrosis. We aimed to determine the active dose and route of administration of Longidaza.
We added in the Supplement: Table S1. Study Design and Experimental Groups, Figure S1 (Effect of Longidaza on perivascular and peribronchial inflammation in the lungs of C57BL/6 mice on d7) and Figure S2 (Content of the connective tissue in the lungs of C57BL/6 mice on d21).
I am fully aware that H and E sections were used historically to evaluate inflammation. Unfortunately, this is no longer considered adequate or precise. The authors have not provided any explanation regarding how they used H and E to look for inflammation, nor are any insets used to indicate what they are focusing on. The authors simply must use unbiased histological methods using specific antibodies.
Response:
H&E is the combination of two histological stains: hematoxylin and eosin. The hematoxylin stains cell nuclei a purplish blue, and eosin stains the extracellular matrix and cytoplasm pink, with other structures taking on different shades, hues, and combinations of these colors. We used the inflammation assessment as method described in the articles [1-6]. This method for assessing inflammation allows us to quickly assess inflammation in the lung tissue of animals. The H&E staining is used to identify the various cell types present in inflammatory infiltrates. At this stage of the drug study we did not study which specific inflammatory cells the drug affects. It was important for us whether the drug had an effect or not. Currently the pathogenesis of pulmonary fibrosis remains unclear, although it has been widely accepted that the abnormal activation of certain inflammatory cells and cytokines in the lung is critical. Histology and hematology were used in the study. In addition, we evaluated the levels of pro-inflammatory mediators (IL-6, TGF-β, TNF-α), hyaluronic acid, hydroxyproline, and collagen type I. In our work we did not aim to investigate the mechanisms of Longidaza action.
Unfortunately, it is not possible to study many of the important mechanisms of drug action in a single study. This was a preliminary study to determine the active dose of Longidaza and the best route of administration. Next, we plan to investigate the mechanism of action of Longidaza and conduct an immunohistochemistry study. We agree with the reviewer that it is very important and interesting.
It was known the H&E staining often is used in articles to assess the effect of drugs, where IHC did not used [7-11].
- Feng L, Sun F, Chen Y, Athari SS, Chen X. Studying the Effects of Vitamin A on the Severity of Allergic Rhinitis and Asthma. Iran J Allergy Asthma Immunol. 2021. 8;20(6):648-692. doi: 10.18502/ijaai.v20i6.8018.
- Ridzuan N, Zakaria N, Widera D, Sheard J, Morimoto M, Kiyokawa H, Mohd Isa SA, Chatar Singh GK, Then KY, Ooi GC, Yahaya BH. Human umbilical cord mesenchymal stem cell-derived extracellular vesicles ameliorate airway inflammation in a rat model of chronic obstructive pulmonary disease (COPD). Stem Cell Res Ther. 2021. 12;12(1):54. doi: 10.1186/s13287-020-02088-6.
- Sheward, D. J., Mandolesi, M., Urgard, E., Kim, C., Hanke, L., Perez Vidakovics, L., Pankow, A., Smith, N. L., Castro Dopico, X., McInerney, G. M., Coquet, J. M., Karlsson Hedestam, G. B., Murrell, B. Beta RBD boost broadens antibody-mediated protection against SARS-CoV-2 variants in animal models. Cell reports. Medicine, 2021. 2(11), 100450. Doi:.10.1016/j.xcrm.2021.100450
- Verma, G., Mukhopadhyay, C. S., Verma, R., Singh, B., Sethi, R. S. Long-term exposures to ethion and endotoxin cause lung inflammation and induce genotoxicity in mice. Cell and Tissue Research. 2018. doi:10.1007/s00441-018-2912-0
- Downing L, Sawarynski KE, Li J, McGonagle M, Sims MD, Marples B. A simple quantitative method for assessing pulmonary damage after x irradiation. Radiat Res. 2010;173(4):536-544. doi:10.1667/RR1712.1
- Zheng, L., Zhu, Q., Xu, C., Li, M., Li, H., Yi, P. Q., Xu, F. F., Cao, L., & Chen, J. Y. Glycyrrhizin mitigates radiation-induced acute lung injury by inhibiting the HMGB1/TLR4 signalling pathway. Journal of cellular and molecular medicine, 2020. 24(1), 214–226. doi:10.1111/jcmm.14703
- Yao, Y., Yuan, Y., Lu, Z., Ma, Y., Xie, Y., Wang, M., Liu, F., Zhu, C., & Lin, C. Effects of Nervilia fordii Extract on Pulmonary Fibrosis Through TGF-β/Smad Signaling Pathway. Frontiers in pharmacology, 2021. 12, 659627. Doi:10.3389/fphar.2021.659627
- Wang, X., Wang, Z., & Tang, D. Aerobic Exercise Alleviates Inflammation, Oxidative Stress, and Apoptosis in Mice with Chronic Obstructive Pulmonary Disease. International journal of chronic obstructive pulmonary disease, 2021. 16, 1369–1379. Doi:10.2147/COPD.S309041
- Gholaminejhad, M., Forouzesh, M., Ebrahimi, B., Mahdavi, S. A., Mirtorabi, S. D., Liaghat, A., Monabati, S. J., Hamza, M. O., Hassanzadeh, G. Formation and activity of NLRP3 inflammasome and histopathological changes in the lung of corpses with COVID-19. Journal of molecular histology, 2022. 53(6), 883–890. Doi:10.1007/s10735-022-10101-w
- Geng, Y., Su, S., Cao, L., Yang, T., Ouyang, W., Liu, L., Wu, B., Zhang, Q., Lu, B., Wang, X. Effect of PD-1 Inhibitor Combined with X-Ray Irradiation on the Inflammatory Microenvironment and Lung Tissue Injury in Mice. Journal of inflammation research, 2022. 15, 545–556. https://doi.org/10.2147/JIR.S350112
- Wu, Z., Wang, X., Yang, R., Liu, Y., Zhao, W., Si, J., Ma, X., Sun, C., Liu, Y., Tan, Y., Liu, W., Zhang, X., DI, C., Wang, Z., Zhang, H., Zhang, Z. Effects of carbon ion beam irradiation on lung injury and pulmonary fibrosis in mice. Experimental and therapeutic medicine, 2013. 5(3), 771–776. Doi:10.3892/etm.2013.881
What is the relationship, if any, between poloxamergaluronidase and Longidaza. Please supply more details outlining specifically what Longidaza is, how this compound was identified, with references. It appears that some of this information is in "Methods". Relevant details need to be included in Introduction.
Response:
In 2011, Bitencourt C.S. and colleagues showed that the hyaluronidase is a novel and promising tool for the treatment of pulmonary fibrosis because hyaluronidases treatment potently blocked bleomycin-induced lung injury and fibrosis while it decreased TGF-β production and collagen deposition. In our publication, the Longidaza is a new formulation of hyaluronidase. We have studied the pharmacological activity of a new drug. We did not study and compare the effect of poloxamer hyaluronidase and Longidaza in animals with bleomycin-induced fibrosis. These are different drugs based on hyaluronidase. We now added information in the Introduction and Methods, and added Figure 1 (Active site view of the in silico visualized hyaluronidase conjugate) to the publication.
Fig 2. Post-hoc test pairwise with "bleomycin control" is used. This may be OK in a sense, but the authors are also also doing a dose response and mode of administration. As multiple variables are being compared, an appropriate test needs to be conducted. I would like to have the dose response and the mode of administration also evaluated. Significance needs to be indicated on the graph itself. The abbreviations im, in and po need to be defined.
Response:
We now have divided Figure 2 and added the information in Figure legend. We included explanations of abbreviations.
New formulation added to the publication.
Figure 3. The effect of Longidaza on the hematological parameters of C57BL/6 mice modeling bleomycin-induced pulmonary fibrosis on d7.
Figure 4. The effect of Longidaza on the hematological parameters of C57BL/6 mice modeling bleomycin-induced pulmonary fibrosis on d21.
Figure 3 and 4 The abbreviations im and in need to be defined. Scale bar needs to be included. That H and E sections were examined needs to be stated in the title, not the last sentence.
Response:
We now have corrected it. We now added information in the Figures legends.
New formulation added to the publication.
Figure 5 (was figure 4) Histological analysis of lung inflammation on d7. Representative photomicrographs of lung tissue sections obtained from male C57BL/6 mice and stained with H&E. x100. Scale bar 50 μm. Groups: Intact - mice of intact control (group 1); Vehicle i.m. and Vehicle i.n. - mice treated with bleomycin (groups 2 and 3); Pirfenidone 300 mg/kg p.o. - pulmonary fibrosis+pirfenidone (group 4); LG 60 U/kg i. m. - pulmonary fibrosis+Longidaza intramuscularly at a dose of 60 U/kg (group 5); LG 120 U/kg i.m. - pulmonary fibrosis+Longidaza intramuscularly at a dose of 120 U/kg (group 6); LG 1200 U/kg i. m. - pulmonary fibrosis+Longidaza intramuscularly at a dose of 1200 U/kg (group 7); LG 10 U/kg i.n. - pulmonary fibrosis+Longidaza intranasally at a dose of 10 U/kg (group 8); LG 30 U/kg i. n. - pulmonary fibrosis+Longidaza intranasal at a dose of 30 U/kg (group 9); LG 120 U/kg i.n. - pulmonary fibrosis+Longidaza intranasal at a dose of 120 U/kg (group 10). P.o. - per os, i.m. - intramuscular, i.n. - intranasal.
Figure 6 (was figure 4). Photomicrographs of lung sections obtained from male C57BL/6 mice on d21. Tissues were stained with H&E. x100. Scale bar 50 μm. Groups: Intact - intact control mice (group 1); Vehicle i.m. and Vehicle i.n. - bleomycin treated mice (groups 2 and 3); Pirfenidone 300 mg/kg p.o. - pulmonary fibrosis+pirfenidone (group 4); LG 60 U/kg i. m. - pulmonary fibrosis+Longidaza intramuscularly at a dose of 60 U/kg (group 5); LG 120 U/kg i.m. - pulmonary fibrosis+Longidaza intramuscularly at a dose of 120 U/kg (group 6); LG 1200 U/kg i. m. - pulmonary fibrosis+Longidaza intramuscularly at a dose of 1200 U/kg (group 7); LG 10 U/kg i.n. - pulmonary fibrosis+Longidaza intranasally at a dose of 10 U/kg (group 8); LG 30 U/kg i. n. - pulmonary fibrosis+Longidaza intranasal at a dose of 30 U/kg (group 9); LG 120 U/kg i.n. - pulmonary fibrosis+Longidaza intranasal at a dose of 120 U/kg (group 10). P.o. - per os, i.m. - intramuscular, i.n. - intranasal.
Both Figures 3 and 4 seem to have identical information. What is the purpose of each figure? What are the conclusions? Please explain in the title and legend. Figure 3 and 4 do not seem to be quantified. Are these representative images? N=? What details are we supposed to look at? Show with arrows, and explain in the legend and title
Response:
We now have corrected it. We now added information in the Figures. Figure 5 (was figure 3) shows photomicrographs of lung where tissues stained with hematoxylin-eosin on d7 of the experiment. Figure 6 (was figure 4) shows photomicrographs of lung where tissues stained with hematoxylin-eosin at d21 of the experiment.
Figure 5 shows a histogram derived from primary data. The primary data needs to be shown on the figure. What inflammatory cells were detected? Macrophage, T cells etc? Show immunohistochemistry detecting individual cell types. Pairwise with bleomycin is a slightly misleading posthoc test, as multiple groups (dose response) is being used. Is there a dose response?
Response:
Staining with hematoxylin and eosin (H&E) is a common technique for staining pathologic tissues, including the lung architecture and inflammatory cells assessment. In our investigation we used this method to assess the lung architecture and inflammatory cells. We understand that our research has limitations. We observed mononuclear infiltration of lung tissue at the bleomycin-induced pulmonary fibrosis. We detected macrophage, T-cells, and other individual cell types. Unfortunately, we did not perform separate cell counts. Additionally, we evaluated the level of pro-inflammatory mediators (IL-6, TGF-β, TNF-α), hyaluronic acid, hydroxyproline, and collagen type I. This was a preliminary study to determine the active dose of Longidaza. Next, we plan to investigate the mechanism of action of Longidaza and conduct immunohistochemistry study.
Figure 6. What is "connectivity tissue". All letters need to be upper left of panel, capitals, no parentheses. I do not understand the purpose of 6B. Include perfinidone in 6C, D. Pairwise with bleomycin is a slightly misleading posthoc test, as multiple groups (dose response) is being used. Is there a dose response?
Response:
We now have corrected Figure. To visualize collagen deposition, we used the Van Gieson method of picrofuchsin staining. Besides, we assessed the collagen type 1 and hydroxyproline in the ELISA assay. Such an approach in researching the connective tissue in lungs is used by many authors [1-6]. We will be using by all means respect all your comments and suggestions in our further investigations.
- Prentø P. Van Gieson's picrofuchsin. The staining mechanisms for collagen and cytoplasm, and an examination of the dye diffusion rate model of differential staining. Histochemistry. 1993 Feb;99(2):163-74.
- Singh M, Chaudhary AK, Pandya S, Debnath S, Singh M, Singh PA, Mehrotra R. Morphometric analysis in potentially malignant head and neck lesions: oral submucous fibrosis. Asian Pac J Cancer Prev. 2010;11(1):257-60.
3 Fukuda C, Goto K, Imamura M, Nakamura T. Bioactive bone cement with a low content of titania particles without postsilanization: effect of filler content on osteoconductivity, mechanical properties, and handling characteristics J Biomed Mater Res B Appl Biomater. 2010 Nov;95(2):407-13. doi: 10.1002/jbm.b.31731.
- Zhan YT, Li L, Weng J, Song X, Yang SQ, An W. Serum autofluorescence, a potential serum marker for the diagnosis of liver fibrosis in rats. Int J Mol Sci. 2012;13(9):12130-9. doi: 10.3390/ijms130912130.
- Hunasgi S, Koneru A, Vanishree M, Shamala R. Keloid: A case report and review of pathophysiology and differences between keloid and hypertrophic scars. J Oral Maxillofac Pathol. 2013 Jan;17(1):116-20. doi: 10.4103/0973-029X.110701.
- Nasrollahi Nia, F., Asadi, A., Zahri, S., & Abdolmaleki, A. (2020). Biosynthesis, characterization and evaluation of the supportive properties and biocompatibility of DBM nanoparticles on a tissue-engineered nerve conduit from decellularized sciatic nerve. Regenerative therapy, 14, 315–321.
Figure 7. See previous comments regarding statistics. Significance is indicated nowhere on the graph.
We now have corrected it. We now added information in the Figures including explanations and indicated significance on the graph
Figure 8. Text and figure legends. How was each marker assessed? ELISA? See previous comments regarding statistics. Significance is indicated nowhere on the graph. Indicate "TGFbeta1" on graph.
Response:
We now have corrected it. We now added information in the Figures.
I am not sure that the authors can comment on side effects of the drug, as I do not see any appropriate assays in this current paper.
Response:
The aim of our study was to investigate the action of Longidaza on the development of bleomycin-induced pulmonary fibrosis, and the effects of Longidaza were compared with the effect of Pirfenidone. We investigated Longidaza at different doses and different routes of administration. This was a screening investigation. We did not observe side effects of the drug.
Unfortunately for us in a single study we did not get to explore many of the important aspects of action of drugs. It is hard to present all the results in a single article. We plan to continue our study.
Round 3
Reviewer 2 Report
I appreciate the authors' responses, and the result is a substantially improved manuscript. However, some issues remain that need to be adequately addressed.
I am fully aware what H and E is and how it was used historically to assess inflammation. The authors, in their Methods, state:
"The degrees of peribronchial and perivascular infiltrates were assessed by the scale of inflammation and quantitative assessment of peribronchial and perivascular mononuclear cells [289]. The drugs were coded and peribronchial inflammation was 261 assessed in a blinded manner using a reproducible scoring system. Each tissue slice was assigned a value from 0 to 3. A value of 0 was taken when no inflammation was detected, a value of 1 was when there was a rare encounter with inflammatory cells, a value of 2 was when most of the bronchi or vessels were surrounded by a thin layer (from one to 2five cells) of inflammatory cells, and a value of 3 was when most of the bronchi and blood vessels were surrounded by a thick layer (more than five cells) of inflammatory cells. Since there were five to seven randomly selected tissue sections per mouse, inflammation scores could be expressed as an average per animal and compared between groups. In addition, cells in the peribronchial and perivascular segments were counted relative to the length of the basement membrane. The total index of bronchial and vascular inflammation was expressed as the number of inflammatory cells/m of the basement membrane."
The authors still have not provided primary data in the form of actual images to illustrate their data, all they have provided are histograms. This is not acceptable. If the authors are wanting to unequivocally assess inflammation, they need to use staining with markers. Such procedures are considered standard practice
I accept the authors opinion that this is preliminary report and that such experimental procedures are beyond the scope of the paper. In this situation, the authors must do the following:
1) provide the primary data supporting the arbitrary scores described in Figure 7 by including, as requested before, insets or equivalent photographically depicting the data for each parameter AND also indicate with arrows specifically what features are being scored. This is necessary so the reviewer can evaluate the validity of the data, and not just rely on a histogram
2) state in the discussion the limitation, so aptly described by the authors in their response, of the data contained in the manuscript namely the inability to ascribe function to any specific immune cell type.
Author Response
We are very grateful to the reviewer for his work in reviewing our manuscript.
We have now revised the manuscript according to the suggestions. Below are the reviewers' comments and our responses. All changes have been incorporated in the revised version.
1) provide the primary data supporting the arbitrary scores described in Figure 7 by including, as requested before, insets or equivalent photographically depicting the data for each parameter AND also indicate with arrows specifically what features are being scored. This is necessary so the reviewer can evaluate the validity of the data, and not just rely on a histogram.
Response:
New formulation added to the publication.
We added in the Supplement: Figure S1. Histology analyses of H&E stained sections.
2) state in the discussion the limitation, so aptly described by the authors in their response, of the data contained in the manuscript namely the inability to ascribe function to any specific immune cell type.
Thanks for the valuable comments. They are very important to us.
New formulation added to the publication:
Our study has some limitations. We performed a preliminary study to determine the active dose of Longidaza and the best route of administration in the bleomycin-induced pulmonary fibrosis. It is not possible to study many of the important mechanisms of drug action in a single study. To assess the mechanism of Longidaza the results presented here are required further validations using immunohistochemistry and cytometry studies etc. It is important to determine which cells are effected by Longidaza.
Round 4
Reviewer 2 Report
The authors have somewhat addressed the question. However, unfortunately, the authors still have not provided the primary data corresponding to Figure 7. They have, however, provided examples of their grading scheme, although they do not indicate which samples correspond to which grading scheme. Although helpful, this information does not address the issue at hand, which is to provide the primary data corresponding to Figure 7.
The authors MUST provide, as a supplemental file as necessary, primary data corresponding to the results of Figure 7, with specific arrows on the figures to direct the reader to the features being scored.
The authors need to present 5 representative panels (intact, vehicle, LG 60, LG 120 and LG 1200) for both the IM and IN groups, and they must precisely indicate the features that allow each group to be scored in a particular fashion.
Author Response
We are very grateful to the reviewer for his work in reviewing our manuscript.
We have now revised the manuscript according to the suggestions. Below are the reviewers' comments and our responses. All changes have been incorporated in the revised version.
1) The authors have somewhat addressed the question. However, unfortunately, the authors still have not provided the primary data corresponding to Figure 7. They have, however, provided examples of their grading scheme, although they do not indicate which samples correspond to which grading scheme. Although helpful, this information does not address the issue at hand, which is to provide the primary data corresponding to Figure 7.
The authors MUST provide, as a supplemental file as necessary, primary data corresponding to the results of Figure 7, with specific arrows on the figures to direct the reader to the features being scored.
The authors need to present 5 representative panels (intact, vehicle, LG 60, LG 120 and LG 1200) for both the IM and IN groups, and they must precisely indicate the features that allow each group to be scored in a particular fashion.provide the primary data supporting the arbitrary scores described in Figure 7 by including, as requested before, insets or equivalent photographically depicting the data for each parameter AND also indicate with arrows specifically what features are being scored. This is necessary so the reviewer can evaluate the validity of the data, and not just rely on a histogram.
Response:
Thanks for the valuable comments. They are very important to us.
Primary data for each mouse is shown in Figure 7. The average data is shown in the Figure S2. We added in the representative panels (intact, vehicle, LG 60, LG 120 and LG 1200) for both the IM and IN groups to Figure 7.
We have used protocol of perivascular and peribronchial inflammation evaluation as described in the articles [1-12].
- Achaiah, A., Rathnapala, A., Pereira, A., Bothwell, H., Dwivedi, K., Barker, R., Iotchkova, V., Benamore, R., Hoyles, R. K., Ho, L. P. Neutrophil lymphocyte ratio as an indicator for disease progression in Idiopathic Pulmonary Fibrosis. BMJ open respiratory research. 2022. 9(1), e001202. doi:10.1136/bmjresp-2022-001202
- Antunes, K. H., Cassão, G., Santos, L. D., Borges, S. G., Poppe, J., Gonçalves, J. B., Nunes, E. D. S., Recacho, G. F., Sousa, V. B., Da Silva, G. S., Mansur, D., Stein, R. T., Pasquali, C., De Souza, A. P. D. Airway Administration of Bacterial Lysate OM-85 Protects Mice Against Respiratory Syncytial Virus Infection. Frontiers in immunology. 2022. 13, 867022. doi:10.3389/fimmu.2022.867022
- Downing L, Sawarynski KE, Li J, McGonagle M, Sims MD, Marples B. A simple quantitative method for assessing pulmonary damage after x irradiation. Radiat Res. 2010. 173(4):536-544. doi:10.1667/RR1712.1
- Feng L, Sun F, Chen Y, Athari SS, Chen X. Studying the Effects of Vitamin A on the Severity of Allergic Rhinitis and Asthma. Iran J Allergy Asthma Immunol. 2021. 8;20(6):648-692. doi: 10.18502/ijaai.v20i6.8018.
- Gueders, M.M.; Bertholet, P.; Perin, F.; Rocks, N.; Maree, R.; Botta, V.; Louis, R.; Foidart, J.M.; Noel, A.; Evrard, B.; et al. A novel formulation of inhaled doxycycline reduces allergen-induced inflammation, hyperrespon: siveness and remodeling by matrix metalloproteinases and cytokines modulation in a mouse model of asthma. Biochem. Pharmacol. 2008. 75, pp. 514–526.
- Ridzuan N, Zakaria N, Widera D, Sheard J, Morimoto M, Kiyokawa H, Mohd Isa SA, Chatar Singh GK, Then KY, Ooi GC, Yahaya BH. Human umbilical cord mesenchymal stem cell-derived extracellular vesicles ameliorate airway inflammation in a rat model of chronic obstructive pulmonary disease (COPD). Stem Cell Res Ther. 2021. 12;12(1):54. doi: 10.1186/s13287-020-02088-6.
- Singh, B., Shinagawa, K., Taube, C., Gelfand, E. W., & Pabst, R. (2005). Strain-specific differences in perivascular inflammation in lungs in two murine models of allergic airway inflammation. Clinical and experimental immunology, 141(2), 223–229. https://doi.org/10.1111/j.1365-2249.2005.02841.x
8.Sheward, D. J., Mandolesi, M., Urgard, E., Kim, C., Hanke, L., Perez Vidakovics, L., Pankow, A., Smith, N. L., Castro Dopico, X., McInerney, G. M., Coquet, J. M., Karlsson Hedestam, G. B., Murrell, B. Beta RBD boost broadens antibody-mediated protection against SARS-CoV-2 variants in animal models. Cell reports. Medicine. 2021. 2(11), 100450. Doi:.10.1016/j.xcrm.2021.100450
- Shinagawa K, Kojima M. Mouse model of airway remodeling: Strain differences. Am J Resp Crit Care Med. 2003;168:959–67.
- Verma, G., Mukhopadhyay, C. S., Verma, R., Singh, B., Sethi, R. S. Long-term exposures to ethion and endotoxin cause lung inflammation and induce genotoxicity in mice. Cell and Tissue Research. 2018. doi:10.1007/s00441-018-2912-0
- Wu, Z., Mehrabi Nasab, E., Arora, P., Athari, S. S. Study effect of probiotics and prebiotics on treatment of OVA-LPS-induced of allergic asthma inflammation and pneumonia by regulating the TLR4/NF-kB signaling pathway. Journal of translational medicine. 2022. 20(1), 130. doi:10.1186/s12967-022-03337-3
- Zheng, L., Zhu, Q., Xu, C., Li, M., Li, H., Yi, P. Q., Xu, F. F., Cao, L., & Chen, J. Y. Glycyrrhizin mitigates radiation-induced acute lung injury by inhibiting the HMGB1/TLR4 signalling pathway. Journal of cellular and molecular medicine, 2020. 24(1), 214–226. doi:10.1111/jcmm.14703
Round 5
Reviewer 2 Report
The authors have supplied the primary data for Figure 7, as requested.